# Adaptive Source Localization on Complex Networks via Conditional Diffusion Model

## Abstract

Network propagation issues like the spread of misinformation, cyber threats, or infrastructure breakdowns are prevalent and have significant societal impacts. Identifying the source of such propagation by analyzing snapshots of affected networks is crucial for managing crises like disease outbreaks and enhancing network security. Traditional methods rely on metrics derived from network topology and are limited to specific propagation models, while deep learning models face the challenge of data scarcity. We propose **ASLDiff** (**A**daptive **S**ource **L**ocalization **Diff**usion Model), a novel adaptive source localization diffusion model to achieve accurate and robust source localization across different network topologies and propagation modes by fusing the principles of information propagation and restructuring the label propagation process within the conditioning module. Our approach can not only capture the characteristics of propagation patterns effectively but also adapt to real-world patterns quickly on synthetic propagation data when domain data is limited. Evaluations of various datasets demonstrate ASLDiff's superior effectiveness, accuracy, and adaptability in real-world applications, showcasing its robust performance across different localization scenarios. The code can be found at https://anonymous.4open.science/r/ASLDiff-4FE0.

## 1 Introduction

In today's highly interconnected world, network propagation issues, such as misinformation spread, cyber threats, and infrastructure failures, have far-reaching consequences for society. The ability to quickly identify the source of these disruptions is critical for mitigating their impact. By analyzing snapshots of affected networks, we can trace the origin of the spread, a process essential for managing crises like disease outbreaks (Ru et al., 2023), enhancing network security (Kephart & White, 1993), and preventing further damage in scenarios such as power grid failures (Amin & Schewe, 2007).

Early methods (Lappas et al., 2010; Shah & Zaman, 2012; Prakash et al., 2012; Luo et al., 2013; Zhu & Ying, 2014a;b) for source localization in networks rely on metrics or heuristics derived from the network's topology, applicable only to specific propagation models like the Susceptible-Infected (SI) or Independent Cascade (IC) models. Notably, Wang et al. (Wang et al., 2017) overcome this limitation by introducing a label propagation algorithm based on the intuition of source prominence (Shah & Zaman, 2011), but still neglect the indeterminacy of information propagation that corresponds to the uncertain nature of source localization. Besides, data-driven methods (Dong et al., 2019; Wang et al., 2022; Hou et al., 2023) are also free from the propagation model limitation as they directly learn a graph neural network (GNN) to capture the propagation process exhibited in empirical data. Recently, deep generative models including variational autoencoders (Ling et al., 2022), normalization flows (Xu et al., 2024) and diffusion models (Huang et al., 2023a; Yan et al., 2024) have been adopted for solving the source localization problem, as they can quantify the indeterminacy in source localization by learning the empirical data distribution and promote the state-of-the-art outcomes.

However, collecting real-world propagation data is difficult and costly, posing significant requirements on source localization models that can adapt to real-world environments with limited data. This brings up two main following challenges. **Firstly, real-world networks typically exhibit unknown propagation patterns, which becomes far more challenging to characterize when data is limited.** In this regard, existing learning-based methods (Dong et al., 2019; Wang et al., 2022; Ling

et al., 2022; Yan et al., 2024) rely purely on data to gain an understanding of the propagation patterns, limiting their capability to generalize in unseen scenarios. **Secondly, complex interrelations between propagation patterns and network topology are difficult to capture with limited data.** Existing deep learning methods rely on a large amount of labeled data from the target network (i.e., identified source nodes from historical propagation) to account for the impact of structural heterogeneity on propagation patterns. However, these models struggle to generalize to new networks when insufficient training data is available.

Therefore, in this paper, we propose a novel method, namely **A**daptive **S**ource **L**ocalization **Diff**usion Model (**ASLDiff**), to achieve accurate and robust source localization across different network topology and propagation patterns, especially under limited real-world data scenarios. Specifically, we propose leveraging the diffusion model (DM) Ho et al. (2020) to tackle the complex source distribution conditioned on the network topology and the current observation of node states for the source localization problem. To address the above two challenges, we enhance the purely data-driven approach by incorporating principles of information propagation—specifically, the prominence of the source and the centrality of rumors—into the design of a conditional diffusion model. First, we propose leveraging pre-calculated source estimations from a label propagation method and using them as informative priors to guide the diffusion and sampling process within the DM framework. This prior knowledge provides consistent guidance when specific information about the propagation pattern is limited. Second, to improve the predictive capability of the denoising network for the source distribution, we enhance it with a conditional input that encodes propagation principles, i.e., the prominence and centrality of nodes in relation to the infected nodes. To obtain this information, we devise a label propagation process and parameterize it using a Graph Convolutional Network (GCN) based architecture, allowing it to better fit empirical data in an inductive learning manner and capture universal propagation patterns across diverse network topologies.

Our contributions are summarized as follows:
(1) We propose a diffusion model-based method ASLDiff for source localization, which effectively learns from simulation and real-world data. ASLDiff effectively captures characteristics of propagation patterns, demonstrating significant practical applicability across diverse scenarios.
(2) We design an innovative conditional diffusion model that incorporates principles of information propagation for improved source distribution prediction. This includes a prior-guided diffusion process and a propagation-enhanced conditional denoiser.
(3) We evaluate the performance of ASLDiff against state-of-the-art methods under various propagation patterns and real network datasets. Additionally, we assess the model's generalizability across different network topologies and propagation patterns, demonstrating its ability to overcome the identified challenges. ASLDiff shows a 7.5%-12.1% improvement in real-world propagation datasets, highlighting its accuracy and adaptability.

## 2 RELATED WORK

### 2.1 SOURCE LOCALIZATION

As the inverse problem of information propagation on networks, source localization refers to inferring the initial propagation sources given the current diffused observation, such as the states of the specified sensors or a snapshot of the whole network status (Shelke & Attar, 2019). It can be applied to tasks like rumor source identification and finding the origin of rolling blackouts in intelligent power grids (Shelke & Attar, 2019). Early approaches focus on single-source identification (Shah & Zaman, 2011; Zhu & Ying, 2014a;b). For example, Shah & Zaman (2011) develop a rumor-centrality-based maximum likelihood estimator under the Susceptible-Infected (SI) (Kermack & McKendrick, 1927) propagation pattern. Later, methods devised for multiple source localization have been proposed (Lappas et al., 2010; Luo et al., 2013; Wang et al., 2017; Dong et al., 2019; Wang et al., 2022). However, most previous approaches fail to model the uncertainty of the location of sources, as the forward propagation process is stochastic. To overcome this, generative models have been adopted. SLVAE (Ling et al., 2022) utilizes the Variational Auto-Encoders (VAEs) backbone and optimizes the posterior for better prediction. However, it is difficult to converge when the propagation pattern is complicated due to the nature of VAEs. DDMSL (Yan et al., 2024) models the Susceptible-Infected-Recovered (SIR) (Kermack & McKendrick, 1927)infection process into the discrete Diffusion Model (DM) (Ho et al., 2020), and design a reversible residual block based

on Graph Convolutional Networks (GCNs) (Kipf & Welling, 2016). However, it requires additional data of the intermediate propagation states and cannot be generalized to other propagation patterns. Our method demonstrates superior functionality and adaptability for real-world applications, requiring fewer input data while addressing existing limitations, thus offering greater practical value. We provide a comparison of typical multiple source localization methods in the Appendix A.

# 3 PRELIMINARIES

## 3.1 PROBLEM FORMULATION

Our research problem is formulated as follows. Given an undirected social network $\mathcal{G} = (V, E)$ where $V$ is the node set, $E$ is the edge set, and $Y = \{Y_1, \ldots, Y_{|V|}\}$ is an infection state of all nodes in $\mathcal{G}$, which describes that a subset of nodes in $\mathcal{G}$ have been infected. Each $Y_i \in \{1, 0\}$ denotes the infection state of node $v_i \in V$, where $Y_i = 1$ indicates that $v_i$ is infected and otherwise $Y_i = 0$ indicates it is uninfected. We aim to find the original propagation source $\hat{X}$ from the propagated observation $Y$, so that the loss with the ground Truth source set $X^* \in \{1, 0\}^{|V| \times 1}$ is minimized, i.e. $\hat{X} = argmin_X ||X - X^*||_2^2$. To account for the uncertainty in source localization, we need to construct a probabilistic model $P(X|Y, G)$, which can be used to sample for the final prediction.

## 3.2 TYPICAL PROPAGATION MODELS

Information propagation estimation involves approximating and reproducing the spread of information in a network and providing explanations based on propagation sources. This task has applications in event prediction (Zhao, 2021), adverse event detection (Wang & Zhao, 2018), and disease spread prediction (Tang et al., 2023). Models for this purpose fall into two main categories: infection models and influence models. Infection models, such as the Susceptible-Infected (SI) and Susceptible-Infected-Susceptible (SIS), manage transitions between susceptible and infected statuses in networks, offering different switching paths for these changes (Kermack & McKendrick, 1927; Keeling & Eames, 2005). Specifically, every infected node attempts to infect adjacent nodes with probability $\beta$ at each iteration. However, in the SIS model, infected nodes might revert to being susceptible with a certain probability $\lambda$. A more complex case is the Susceptible-Infected-Recovered (SIR) model, which additionally considers the recovered state.

Independent Cascade (IC) and Linear Threshold (LT) (Kempe et al., 2003) are two typical influence models examining how influence spreads in social networks or infrastructure networks. The IC model involves nodes that can either be active or inactive. The process begins with a set of initial active nodes. At each step, any newly activated node can activate its inactive neighbors with a single chance. The probability of activation is dependent on the weight of the edge between nodes. As for the LT model, each inactive node becomes active only if it receives enough influence (over a threshold) from its neighbors.

## 3.3 LABEL PROPAGATION BASED SOURCE IDENTIFICATION

In realistic situations, the intractable propagation process does not have an explicit prior, and it is also challenging to value appropriate parameters for the pre-selected underlying propagation model. To address this, Wang et al. (2017) introduce source prominence and centrality characteristics in the method design. The former comes from the common observation that sources are surrounded by more infected nodes, while the centrality of sources shows that nodes far from the source are less likely to be infected than those near it (Shah & Zaman, 2012), which can also be observed in the real-world data by our analysis in the Appendix B. Based on these ideas, they propose to perform label propagations on the observation state of the network. By setting $Y[Y = 0] = -1$ and $\mathcal{Z}^{t=0} \leftarrow Y$, the iteration of label propagation and the convergence states are as follows:

$$\mathcal{Z}_i^{t+1} = \alpha \sum_{j:j \in \mathcal{N}(I)} S_{ij} \mathcal{Z}_j^t + (1 - \alpha) Y_i. \tag{1}$$

$\mathcal{Z}$ finally converges to: $\mathcal{Z}^* = (1 - \alpha)(I - \alpha S)^{-1} Y$, where $S = D^{-1/2} A D^{-1/2}$ is the normalized weight matrix of graph $\mathcal{G}$, $\alpha$ is the fraction of label information from neighbors, and $\mathcal{N}(i)$ stands for the neighbor set of the node $i$. After obtaining the converged matrix $\mathcal{Z}^*$, one node can identified as a source when its final label is larger than its neighbors.

## 4 ASLDiff: THE PROPOSED METHOD

In this section, we demonstrate our proposed diffusion model for adaptive source localization. The overall framework of this model is presented in Figure 1. Specifically, we propose to leverage the advice of the pre-calculated estimation of the source from the label propagation approach and treat it as an informative *prior* to guide the diffusion and sampling process in the DM framework. Moreover, we devise the denoising network $f_\theta$ and employ a GCN-based conditional module to extract the message of the nodes' prominence and centrality among the infected subgraph, and learn the invariant features of the propagation pattern across diverse network topologies.

### 4.1 PRIOR-GUIDED DIFFUSION PROCESS

To capture the indeterminacy of the ill-posed localization problem, it is essential to build a probabilistic model that can also leverage the topological information in the graph structure. We consider using the generative DM framework to tackle this challenge by modifying it as a source predictor, which classifies each node into two categories: source or non-source. In the training process, the DM gradually introduces noise into data and then learns to reverse this process by training the denoising network. It gradually transforms pure Gaussian noise into the original data, generating new samples as source predictions from the learned distribution. However, as the network grows, it becomes harder to estimate the

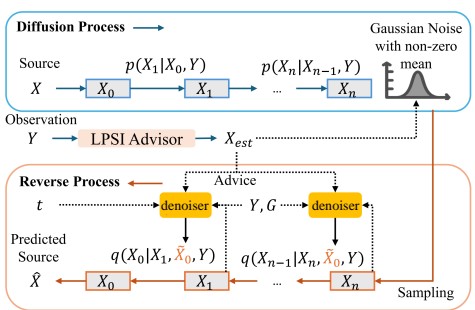

Figure 1: The framework of ASLDiff.

sources' location due to the increase in the distributional space of the source vector. However, the vanilla diffusion models assume the same endpoint of the diffusion process. In other words, the generation process for all regions starts from the same Gaussian noise $\mathcal{N}(0, I)$, which makes it difficult to recover the label simply from its conditional observation inputs $Y$. According to Ali et al. (2020), classical non-deep learning methods still provide reasonable predictions for source localization. Therefore, to enhance DM's effectiveness and efficiency, we propose leveraging pre-calculated source estimations as the advice from the label propagation-based source identification method and using them as informative priors to guide the diffusion and sampling process within the DM framework to reduce data fitting difficulty and improve efficiency and effectiveness. Specifically, we treat the estimation $X_{est} \in \{0, 1\}^{|V| \times 1}$ as a soft-label vector of sources to guide the forward diffusion and reverse process of our diffusion generation framework. The soft-label is calculated using the converged form of Equation (1). On the one hand, it is treated as a condition of the denoising network. On the other hand, inspired by Han et al. (2022), we modify the mean of the diffusion endpoint as the soft label $X_{est}$ to incorporate domain knowledge about source characteristics for each input $Y$, instead of using standard Gaussian noise, allowing our model to leverage reliable prior knowledge while maintaining the flexibility to explore the full solution space through the denoising process.

Specifically, in the diffusion process, our DM framework incrementally corrupts the source label $X = X_0$ into the Gaussian noise via a Markov chain:

$$p(X_{1:n}|X_0, Y) = \prod_{t=1}^{n} p(X_t|X_{t-1}, Y) \tag{2}$$

leading to the endpoint of the diffusion process to be:

$$p(X_n|Y) = \mathcal{N}(X_{est}(Y), I). \tag{3}$$

According to the original notation in Ho et al. (2020), the Markov transition can be modified as:

$$p(X_t|X_{t-1}, Y) = \mathcal{N}(\sqrt{1 - \beta_t}X_{t-1} + (1 - \sqrt{1 - \beta_t})X_{est}, \beta_t I), \tag{4}$$

which derives the closed-form distribution with arbitrary $t$:

$$p(X_t|X_0, Y) = \mathcal{N}(\sqrt{\bar{\alpha}_t}X_0 + (1 - \sqrt{\bar{\alpha}_t})X_{est}, (1 - \bar{\alpha}_t)I), \tag{5}$$

where $\{\beta_t\}_{0:n} \in (0,1)^n$ is a predefined diffusion schedule and $\alpha_t := 1 - \beta_t, \bar{\alpha}_t := \prod_t \alpha_t$. Properly choosing the schedule and the maximum diffusion timestep $n$ will make the endpoint $(X_n|Y)$ close enough to our instruction above.

Besides, in the reverse denoising process, we aim to build a reverse Markov denoiser $p_\theta(X_{t-1}|X_t, Y) = p(X_{t-1}|X_t, Y, f_\theta)$ to recover the original data. DM framework trains the parameterized denoiser to fit the ground truth posterior:

$$q(X_{t-1}|X_t, X_0, Y) = \mathcal{N}(\tilde{\mu}(X_t, X_0, Y), \tilde{\beta}_t I), \tag{6}$$

where

$$\tilde{\mu}(X_t, X_0, Y) := \frac{\sqrt{\bar{\alpha}_{t-1}}\beta_t}{1 - \bar{\alpha}_t}X_0 + \frac{(1 - \bar{\alpha}_{t-1})\sqrt{\alpha_t}}{1 - \bar{\alpha}_t}X_t$$

$$+ (1 + \frac{(\sqrt{\bar{\alpha}_t} - 1)(\sqrt{\alpha_t} + \sqrt{\bar{\alpha}_{t-1}})}{1 - \bar{\alpha}_t})X_{est}(Y), \tag{7}$$

$$\tilde{\beta}_t := \frac{1 - \bar{\alpha}_{t-1}}{1 - \bar{\alpha}_t}\beta_t.$$

The core of the denoiser is the denoise network $f_\theta$, and is set to estimate the ground truth source $X(i.e. X_0)$, which we empirically find more effective. In other words, the parameterized denoising network $f_\theta$ is trained to fit $X_0$ in the Equation (7). The denoise network outputs the estimated source vector $\tilde{X}_0 := f_\theta(X_t, X_{est}, Y, \mathcal{G}, t)$ to calculate the posterior for step-by-step denoising. The denoising network $f_\theta$ can be trained by the simple L2 loss function:

$$L(\theta) = \mathrm{E}_{X_0 \sim p(X_0|\cdot), t}||X - f_\theta(X_t, t, \cdot)||_2^2 \tag{8}$$

The above-mentioned framework is illustrated in Figure 1.

### 4.2 PROPAGATION-ENHANCED CONDITIONAL DENOISER

In this section, we introduce the denoising network parameterization enhanced by label propagation, which is an effective infusion of the prominence and centrality principle of sources. The observation input is encoded via label propagation, analogous to message-passing in graphs. To better capture universal propagation patterns, we propose using a Graph Convolutional Network to parameterize the label propagation process in Equation (1).

#### 4.2.1 DENOISING NETWORK ARCHITECTURE

The architecture of our denoising network is shown in Figure 2.

**Encoding the noisy input and soft labels.** The soft-label $X_{est}$ is forwarded through a multi-layer GNN to capture the hidden message with graph structural information. Subsequently, it is added to the noisy input $X_t$ and passed through a linear layer. The final input for the GNN encoder is $Z_e = \mathrm{Linear}(\mathrm{GNN}(X_{est}) \oplus X_t) \oplus Emb(t)$, where for the denoising step t, we use the classical sinusoidal embedding (Vaswani et al., 2017). The $\oplus$ indicates element-wise sum. $Z_e$ is then passed through a GCN-based encoder and is smoothed through a softmax function $\sigma$ and layer normalization:

$$Z_d = \mathrm{LN}(\sigma(\mathrm{GNN}(Z_e))).$$

Softmax and layer normalization operations are then used to improve the network's representational capacity and convergence performance, resulting in better performance and faster training (Huang et al., 2023b).

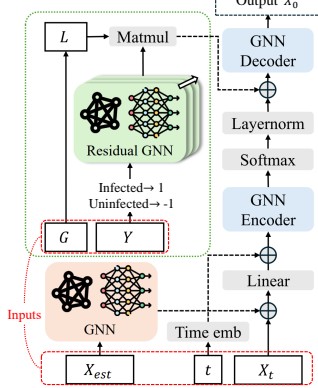

Figure 2: The architecture of the denoising network.

**Conditioning.** Shown at the left part of the figure, a GCN-based module learns the encoding carrying the source prominence and centrality from the infection state input $Y$, which will be elaborated on in the next section.

**Decoder.** $Z_d$ and encoded condition $h_{out}$ are decoded through a GCN-based module, resulting in the estimation for the uncorrupted sample $X_0$ (i.e. $X$):

$$\tilde{X}_0 = \mathrm{GNN}(Z_d, h_{out}).$$

### 4.2.2 DENOISING NETWORK CONDITIONING DESIGN

Our conditioning module takes the observed infection states as input. Considering leveraging the previously described empirical knowledge of source nodes, we aim for this module to extract effective encoding information from the infection states that represents the degree of prominence and centrality for each infected node. A straightforward approach to achieve this is through direct label propagation (Wang et al., 2017), which firstly labels the infected or influenced nodes in a network as the positive integer 1, while labeling the other nodes as -1. By propagating these labels throughout the network, the features of proximity and centrality are captured. However, the rigid and homogeneous nature of this propagation process lacks the requisite flexibility and adaptive learning capabilities necessary for optimal performance across diverse network scenarios.

To better utilize the graph structure and extract hidden messages of the propagation pattern from data, we adopt GNNs to parameterize the label propagation process and generate more informative conditional features. In Equation (1), the label of a node in the next step is a combination of its original label and the sum of normalized labels from its neighbors. We can rewrite this iteration as:

$$\mathcal{Z}_i^{t+1} = \hat{\alpha} Y_i + \sigma(\sum_{j:j\in\mathcal{N}(I)} \phi(\mathcal{Z}_j^t, S_{ij})), \tag{9}$$

where we add non-linear transformations $h(\cdot)$ and $\sigma(\cdot)$ to enhance the expressiveness of the propagation process. The structure of the above equation exactly matches the form of the general Graph Neural Network (GNN) (Gilmer et al., 2017), and can be achieved by using a residual block combined with a graph convolutional network(GCN, Kipf & Welling (2016)):

$$Y[Y = 0] = -1, \ h^{(0)} = YU^T, U \in \mathrm{R}^{C\times 1},$$
$$g(h^{(l)}) = \sigma(\tilde{D}^{-1/2}\tilde{A}\tilde{D}^{-1/2} \cdot h^{(l)} \cdot w), \ h^{(l+1)} = h^{(0)} + g(h^{(l)}). \tag{10}$$

Among them, $U$ is the linear transformation, $\sigma$ is the activation operator PReLU, $h^{(l)}$ stands for the output hidden state of the $l$-th layer of the GCN, $\tilde{A} = A + I$ is the adjacency matrix with self-loops, and $\tilde{D}$ is the degree matrix of $\tilde{A}$. The final layer's output $h^{(l_f)}$ is projected back to dimension 1 and multiplied by the graph's Laplacian matrix $L$, i.e. $h_{out} := L \cdot h^{(l_f)}$. The GCN structure allows the model to learn adaptive propagation rules by combining fixed theoretical principles (encoded in label propagation) with data-driven features. $h_{out}$ is then added to the latent embedding from the encoder, as shown in Figure 2.

Enabled by our prior-guided diffusion process and propagation-enhanced conditioning design, our model is enhanced by universal knowledge across propagation patterns: source prominence and centrality. Two benefits can be obtained: (1) when sufficient domain data is available, it can help the model capture characteristic of propagation pattern more effectively. Our model can be directly trained on domain datasets; (2) when domain data is limited, the model can be pretrained on synthetic propagation data simulated on established propagation models and perform efficient few-shot or zero-shot learning. This is because our model can effectively learn pattern-invariant features from pretrain data under the enhancement of knowledge, which is more practical in real-world cases.

## 5 EXPERIMENTS

For this study, we utilize real-world datasets to evaluate our proposed model for answering the following questions:

**Q1. Accuracy:** How does ASLDiff perform compared with other source localization methods under different diffusion patterns (e.g., SIS, IC, Real-world Scenarios)? (In this part, the training and testing are performed on the same dataset.)

**Q2. Adaptability**: How well does ASLDiff perform on real-world network topologies/propagation patterns after trained/pretrained on synthetic networks/patterns? (In this part, the few-shot and zero-shot capability of ASLDiff is validated.)

**Q3. Ablation Study**: How does each component of ASLDiff contribute to the overall system?

Table 1: Performance under SIS diffusion pattern. The best performance is indicated in bold, and the second-best performance is indicated with underline.

| Dataset | | **Net** | | | | **Jazz** | | | | **Power** | | | |
|---------|--------|-------|-------|-------|-------|-------|-------|-------|-------|-------|-------|-------|-------|
| Type | Method | F1 | RE | PR | AC | F1 | RE | PR | AC | F1 | RE | PR | AC |
| Rule-based | Netsleuth | 0.523 | 0.519 | 0.526 | 0.952 | 0.018 | 0.017 | 0.019 | 0.915 | 0.606 | 0.605 | 0.608 | 0.960 |
| | LPSI | 0.717 | 0.926 | 0.585 | 0.966 | 0.153 | 0.963 | 0.083 | 0.535 | 0.738 | 0.911 | 0.619 | 0.968 |
| DL-based | GCNSI | 0.761 | 0.862 | 0.681 | 0.970 | 0.613 | 0.615 | 0.610 | 0.980 | 0.843 | 0.833 | 0.854 | 0.984 |
| | SLVAE | 0.764 | **0.987** | 0.624 | 0.969 | **0.750** | **1.000** | 0.600 | 0.970 | 0.759 | **0.975** | 0.621 | 0.970 |
| | TGASI | 0.781 | 0.922 | 0.676 | 0.971 | 0.672 | 0.740 | 0.613 | 0.980 | 0.849 | 0.805 | 0.898 | 0.984 |
| | DDMSL | 0.801 | 0.930 | 0.702 | 0.979 | 0.708 | 0.844 | 0.609 | 0.977 | 0.767 | 0.966 | 0.636 | 0.980 |
| Ours | Ours | **0.816** | 0.932 | **0.726** | **0.979** | 0.720 | 0.838 | **0.635** | **0.980** | **0.877** | 0.854 | **0.902** | **0.985** |

## 5.1 Experiment Settings

### 5.1.1 Datasets

Following Ling et al. (2022); Yan et al. (2024), we use both synthetic and real-world propagation data to evaluate ASLDiff. For the synthetic dataset, we select three real-world networks that may be involved in disease or message propagation: *network science* (Net),*jazz*, and *power grid* (Power). We simultaneously use the SIS, SIR, IC and LT forward propagation models to simulate 100 steps or until convergence, thus obtaining multiple sets of synthetic data. For real-world datasets *Digg* and *Twitter*, which both have more than 10000 nodes, the real propagation cascades are available. For each cascade in both sets, we designate the infected nodes at the first 10% of the propagation time as source nodes and take the network's infection status at 30% of the propagation time as observation input. In the context of real-world applications, we often can only collect sufficient data for analysis after some time has elapsed since the occurrence of the event. Therefore, attempting to predict what initially happened in the process when we have observed enough propagation patterns at a certain degree of infection time is very much in line with the needs of real-world operations. Please refer to the Appendix D for specific details of the datasets.

### 5.1.2 Baselines, Experimental Settings, and Metrics

Following previous works (Ling et al., 2022; Yan et al., 2024), we selected two representative heuristic methods, *i.e.*, Netsleuth (Prakash et al., 2012) and LPSI (Wang et al., 2017), and deep learning methods, *i.e.*, GCNSI (Dong et al., 2019), SLVAE (Ling et al., 2022), TGASI (Hou et al., 2023) and DDMSL (Yan et al., 2024). These baselines are all state-of-the-art (SOTA) multi-source localization methods in their domains. Please refer to the Appendix E for specific implementations of baselines and our method.

Following previous works (Wang et al., 2022), we adopt four metrics: 1) F1-score (F1): The harmonic mean of recall and precision, emphasizing the balance between precision and recall; 2) Recall (RE): The proportion of positive cases (source nodes) that are correctly identified, focusing on the model's ability to detect all relevant instances; 3) Precision (PR): The proportion of actual positive cases among the samples judged as positive, highlighting the model's ability to avoid false positives; 4) Accuracy (AC): The proportion of correctly classified nodes, offering an overall measure of correct predictions across all classes.

## 5.2 Accuracy

### 5.2.1 Performance on Synthetic Datasets

The experimental performance comparison of various methods under the datasets of different networks synthesized by the SIS propagation model is shown in Table 1. It shows that our method outperforms the baseline in most metrics, achieving competitive results. Specifically, our proposed method outperforms the existing methods in the accuracy metric on all datasets and performs better on the Net and the Power dataset, with F1 improvement of 1.8-3.2% compared to other baselines. Deep learning-based methods show close performance against our method. Compared to the best performing DL-based method, SLVAE, which achieves the best recalls among three datasets, our

Table 2: Performance under IC diffusion pattern. The best performance is indicated in bold, and the second-best performance is indicated with underline.

| Dataset | | **Net** | | | | **Jazz** | | | | **Power** | | | |
|---|---|---|---|---|---|---|---|---|---|---|---|---|---|
| Type | Method | F1 | RE | PR | AC | F1 | RE | PR | AC | F1 | RE | PR | AC |
| Rule-based | Netsleuth | 0.018 | 0.009 | 0.384 | 0.949 | 0.184 | 0.102 | **0.923** | 0.958 | 0.006 | 0.003 | 0.384 | 0.949 |
| | LPSI | 0.446 | 0.406 | **0.495** | 0.949 | 0.873 | **1.000** | 0.775 | 0.980 | 0.493 | 0.487 | 0.498 | 0.949 |
| DL-based | GCNSI | 0.033 | 0.018 | 0.346 | 0.947 | 0.136 | 0.077 | 0.600 | 0.955 | 0.300 | 0.198 | **0.618** | 0.943 |
| | TGASI | 0.392 | 0.402 | 0.383 | 0.949 | 0.771 | 0.753 | 0.790 | 0.971 | 0.343 | 0.239 | 0.611 | 0.944 |
| Ours | Ours | **0.480** | **0.484** | 0.477 | **0.950** | **0.901** | 0.966 | 0.849 | **0.984** | **0.516** | **0.515** | 0.507 | **0.950** |

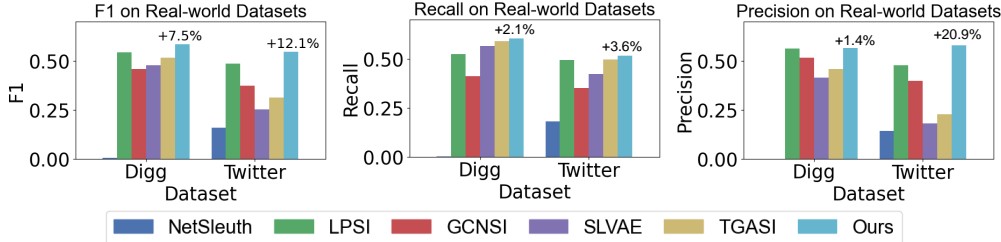

Figure 3: Performance (F1-score, Recall, Precision) on real-world datasets *Digg* and *Twitter*.

method consistently performs better precision, achieving improvement of up to 45%. This is particularly valuable since misidentifying source nodes (false positives) is costly in practical applications, since resources would be wasted investigating non-source nodes. As the best in the Jazz dataset, SLVAE only recalls 1-2 more source nodes compared to our method, but it also produces 3-4 more false positives, since only around 10 nodes are chosen as the ground truth sources in each infection. Accuracy (AC) measures overall classification correctness across all nodes and can be misleadingly high due to the large class imbalance (very few nodes are actual sources). Therefore, the balanced metric F1 should be considered the more critical metric in source localization problems. Considering F1 and the above analysis, our gap between SLVAE in Jazz is not significant, and ASLDiff shows greater superiority against SLVAE in the other two datasets. In all, ASLDiff's better performance in terms of the F1 score across these datasets more effectively demonstrates its superiority.

Table 2 shows the performance comparison of various methods on datasets generated by the IC propagation model, where our method generally surpasses the baseline in most metrics. Our method consistently achieves optimal or near-optimal results across all metrics. Netsleuth underperforms due to its specific design for SI/SIR patterns. GCNSI, on the other hand, shows low recall in this model, though its precision and accuracy are somewhat better, suggesting it detects fewer source nodes than it should. Unlike GCNSI, our method effectively learns the distribution of source nodes, improving both precision and recall, thus achieving higher F1 scores compared to all baselines. The SLVAE model is excluded due to non-converging training on the IC dataset, highlighting issues like difficult training and posterior collapse in VAEs. DDMSL requires calculating the state transfer matrix, which is fundamentally based on the formula of the SIR propagation model and is inapplicable to IC. ASLDiff outperforms the other deep learning baselines according to the F1 score, with at most 72% improvement, indicating our superior generalizability against pure data-driven methods.

Due to the space limits, we present the experimental results of LT and SIR patterns in the Appendix C.

### 5.2.2 PERFORMANCE ON REAL-WORLD DATASETS

The experimental results under real-world propagation patterns are shown in Figure 3. We compare ASLDiff with the above baseline methods except for DDMSL, which is based on the SIR propagation model's framework. Our method consistently exhibits the best or second-best performance across all metrics, with the highest F1 score, demonstrating our effectiveness in larger networks and real-world scenarios. This superior performance is partly due to the conditioning design that encodes propagation principles, enabling our model to achieve performance comparable to LPSI.

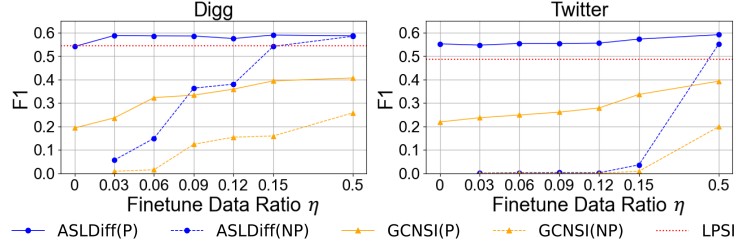

Figure 4: Model's adaptability in terms of propagation patterns. "P" ("NP") stands for pre-training (or not) model using simulation data (SIS and IC). The model is then tested on real-world propagation data (*Digg* and *Twitter*) under both zero-shot (no fine-tuning data, $\eta = 0$) and few-shot ($\eta \in (0, 0.5]$) settings.

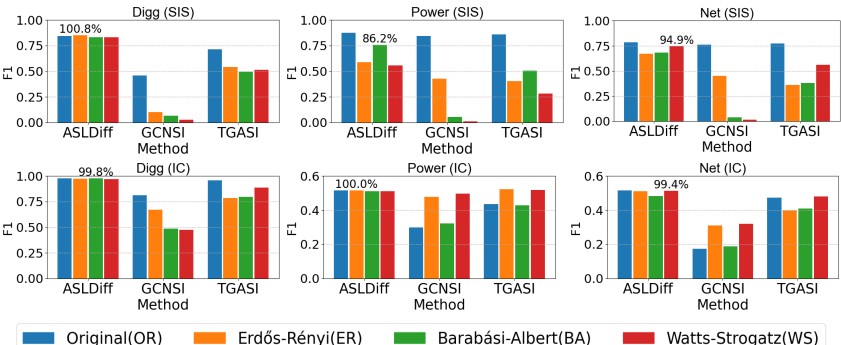

Figure 5: Model's adaptability in terms of network topology. "ER", "BA" and "WS" stand for training on corresponding synthetic networks and test on real-world networks (i.e., *Digg* and *Twitter*). "OR" directly borrows the F1 reported in Table 1-2. Both SIS and IC propagation patterns are considered. For ASLDiff, the ratio between the best of ("ER", "BA", "WS") and "OR" is reported.

Additionally, applying a data-driven generative framework allows our model to capture the complex distribution of sources in real-world scenarios, fully utilizing the internal correlations of the data to grasp more critical distribution characteristics.

### 5.3 ADAPTABILITY

#### 5.3.1 TRANSFER FROM SIMULATED PROPAGATION PATTERN

To validate our method's transferability and few-shot/zero-shot learning capabilities in real-world scenarios, we simulate pre-training data using established propagation models (IC+SIS) within actual networks. ASLDiff undergoes pre-training and is fine-tuned on real propagation pattern data, which effectively addresses the scarcity of real data by utilizing simulation data. Figures 4 present the results on the Digg and Twitter datasets, respectively, compared with baseline GCNSI and LPSI, including no pre-training condition. These methods identify the sources without using the information of the underlying propagation model, which is appropriate for comparison. In the Digg dataset, ASLDiff, pre-trained on simulated data, requires only 3% of the real dataset for fine-tuning to achieve optimal performance, while models without pre-training need about 50%, demonstrating our model's few-shot capability. In the Twitter dataset, our pre-trained model can even reach optimal performance without additional fine-tuning, demonstrating our model's zero-shot capability. In contrast, GCNSI cannot surpass pre-trained ASLDiff with any amount of fine-tuning data in both datasets. This is attributed to the fact that GCNSI simply inputs designed features into the GNN model, which is insufficient to enable it to capture the general distribution laws of different propagation patterns. LPSI, as a non-learning method, is incapable of learning superior features and patterns from the simulation data and thus fails to surpass the pretrained ASLDiff.

#### 5.3.2 TRANSFER FROM SYNTHETIC NETWORKS

To validate the generalization ability of ASLDiff across different network topologies, we generate multiple random networks using classical network generation algorithms: Erdős-Rényi (ERDdS & R&wi, 1959), Barabási–Albert (Barabási & Albert, 1999), and Watts-Strogatz (Watts & Strogatz,

1998). Propagation samples are then produced using propagation models (SIS, IC), with detailed generation procedures provided in the Appendix D.4. The training is conducted entirely independently of the target network. However, when testing the model, the topology of the target network is known to the model. We compare with the DL-based methods from the baselines capable of cross-network transfer experiments, GCNSI and TGASI. The purpose of this experimental design is to validate the model's zero-shot capabilities on new networks. It also aims to demonstrate that when a real-world network lacks sufficient historical propagation data for training, our pre-trained model on synthetic networks can be directly applied for source localization within that network. Models trained separately on datasets generated from these three algorithms are tested on real networks (Digg, Power, Net) with the same corresponding propagation patterns. When real network data under practical applications are unavailable, training on a wide variety of random networks with the same propagation patterns helps the model recognize universal rules of source localization across different networks, as shown by the results presented in Figure 5.

Our model trained on synthetic networks performs very closely to the one originally trained on real networks (as shown in the percentage above the ASLDiff bar for the best-performing synthetic training data, which indicates the relative performance compared to the original network). In contrast, the baselines fail to achieve the same performance on most synthetic datasets as those trained on real networks, nor does it surpass our method. This may be partly due to our diffusion-based distribution learning framework, which enables the model to capture the distribution patterns of sources across different networks from a distributional perspective. Additionally, our parameterized GCN-based propagation-enhanced conditional denoiser, where our model fits empirical data in an inductive learning manner and captures universal propagation patterns across diverse network topologies. Overall, ASLDiff's adaptability across networks is validated.

## 5.4 Ablation Study

We then perform the ablation study for ASLDiff to investigate the importance of prior guided diffusion (PGD) and propagation-enhanced conditioning of the denoiser (PCD). For the first ablated model, we downgrade the advised diffusion process into the original version by setting the endpoint back to be $\mathcal{N}(0, I)$. Hence the reverse sampling must start from a non-guided Gaussian white noise.

For the second ablated version, the conditioning module is replaced by a simple Multi-Layer Perceptron (MLP) with a comparable number of parameters. We evaluate these ablations and compare them to our model in Figure 6. Overall, the performance degrades obviously when our model is ablated, demonstrating our effectiveness. Moreover, removing the conditioning module leads to more significant deterioration in some datasets, indicating the importance of devising the operation process of conditional observation input in the denoising network.

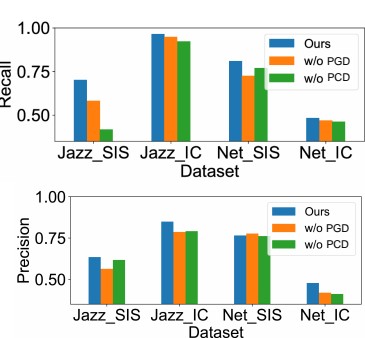

Figure 6: Test results of the ablation study.

## 6 Conclusion

In this paper, we proposed a diffusion model-based method for source localization in complex networks, leveraging GNNs to enhance the model's adaptability to diverse network topologies.By incorporating soft labels and a restructured label propagation process, ASLDiff effectively captures essential propagation characteristics across various network topologies, and is able to quickly adapt to unseen propagation patterns with limited fine-tuning real-world data. Extensive experiments on multiple datasets demonstrate ASLDiff's superior accuracy, efficiency, and generalizability compared to state-of-the-art methods. This work highlights the importance of adaptive capacities in deep learning models for solving the inverse problem of graph diffusion, with significant implications for controlling the spread of diseases, rumors, and other critical societal issues.

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

## A  COMPARISON OF MULTIPLE SOURCE LOCALIZATION METHODS

Table 3: Comparison of different source localization methods. **Ind.**: Indeterminacy. **Obs. input**: Observation input. **ZS**: Zero-shot inference on real-world data (trained on synthetic data). **KI**: Knowledge-informed

| Category | Method | Ind. | Applicable patterns | Obs. input | ZS | KI |
|---|---|---|---|---|---|---|
| Rule-based | NetSleuth((Prakash et al., 2012)) | ✗ | SI | single snapshot | - | - |
| | OJC((Zhu et al., 2017)) | ✗ | SI, SIR, IC | single snapshot | - | - |
| | LPSI((Wang et al., 2017)) | ✗ | SI, SIR, IC | single snapshot | - | - |
| Data-driven | GCNSI((Dong et al., 2019)) | ✗ | SI, SIR, IC | single snapshot | ✓ | ✓ |
| | IVGD((Wang et al., 2022)) | ✗ | IC | single snapshot | ✗ | ✗ |
| | SLVAE((Ling et al., 2022)) | ✓ | SI, SIR, real-world | single snapshot | ✗ | ✗ |
| | SLDiff((Huang et al., 2023a)) | ✗ | real-world | multiple snapshot | ✗ | ✗ |
| | TGASI((Hou et al., 2023)) | ✗ | SI, SIR, IC | multiple snapshot | ✗ | ✗ |
| | DDMSL((Yan et al., 2024)) | ✓ | SI, SIR, real-world | single snapshot | ✗ | ✓ |
| | PGSL((Xu et al., 2024)) | ✓ | SI, SIR, real-world | single snapshot | ✗ | ✗ |
| | GINSD((Cheng et al., 2024)) | ✗ | IC | single snapshot | ✗ | ✗ |
| | Ours | ✓ | SI(S/R), IC, real-world | single snapshot | ✓ | ✓ |

In the above table, we compare the functionality, requirements, and application scenarios of mainstream source localization methods. "**Ind.**" refers to whether the method considers modeling the indeterminacy of source locations. "**Applicable patterns**" refers to the specific propagation pattern to which the method can be applied. "**Obs. input**" refers to the required input for the method to detect the sources. "**ZS**" refers to whether the data-driven method can perform zero-shot inference on real-world data, after trained on synthetic data. "**KI**" refers to whether the data-driven method is knowledge-informed.

From the demonstration, we can observe that our method is designed to be the most functional and capable of handling a broader range of real-world applications. Our method also requires less input data, which is more practical. The method proposed holds significant practical value and addresses the limitations of the existing methods. Additionally, as another method based on the diffusion model, DDMSL and TGASI require the propagation process data during training and the acquisition or calculation of parameters for the infectious model before source localization. This limitation restricts the model's practical application value. Also, PGSL resembles SLVAE's framework and merely utilizes a flow-based model to replace the VAE in SLVAE, while our diffusion model exhibits stronger distribution modeling capabilities. GINSD considers incomplete user data scenarios and utilizes a positional embedding module to distinguish incomplete nodes in the source inference process, and as we do not consider such circumstances, GINSD reduces to a simple GAT-based baseline similar to GCNSI.

It should also be noted that two recent works (Wang et al.; Ling et al., 2024) focus on source localization in a cross-platform setting, which is orthogonal to our research problem and thus not discussed.

## B  ANALYZING SOURCE CENTRALITY IN EMPIRICAL DATA

We believe that the source centrality assumption is not only common in most existing propagation patterns and real-world scenarios, as evidenced by the literature (Ali et al., 2020; Dong et al., 2019), but also validated by the competitive performance on real-world datasets of baselines like LPSI and GCNSI, which are devised based on similar assumptions. We show the analytical results demonstrating the effectiveness of the assumption in the following.

In our analysis of the real-world dataset Digg, we evaluate the normalized(max-min) closeness centrality density and frequency of the source nodes in the subgraph consisting of infected nodes to partially reflect the centrality characteristic of the sources. The closeness centrality (CC) specifically reflects the node topological distance to all other nodes in the subgraph, rather than its degree attribute. The result is shown in Figure 7(a). From Figure 7(a), the mean normalized closeness centrality of sources is higher than the average of all infected nodes, and source nodes cover over 63% of the nodes with the centrality score exceeding 0.8, as shown in Figure 7(b). The overall results

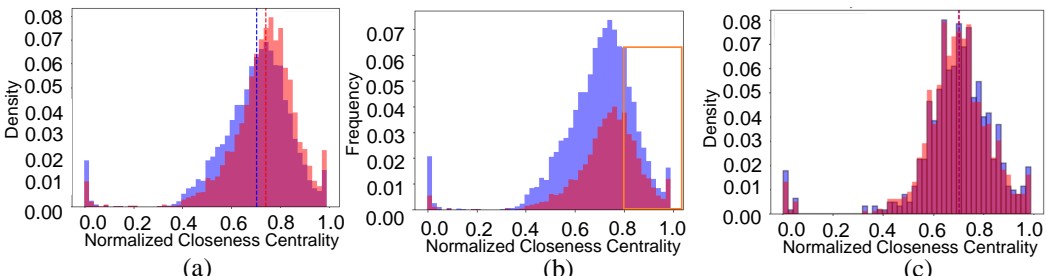

Figure 7: (a)(b) Normalized closeness centrality of infected users and sources for all cascades in *Digg*. The blue histogram shows the normalized closeness centrality distribution of infected nodes, while the red one shows that of source nodes. (a) is the density distribution of closeness centrality. The dashed line indicates the mean centrality for each node type. (b) is the frequency distribution of closeness centrality. The orange box highlights the part where centrality is above 0.8. (c) The closeness centrality(CC) probability density function of the predicted and ground truth source nodes on the *Digg* dataset. The blue histogram shows the normalized closeness centrality distribution of the ground truth sources, while the red one shows that of the predicted ones.

demonstrate the crucial role of source nodes in the information diffusion process and their higher likelihood of being central to the network structure within the cascades.

Our proposed method, ASLDiff, rather than strictly adhering to the centrality in outputting prediction results, exhibits stronger expressive capabilities. Intuitively, on homogeneous networks—where the probability of propagation along network edges is the same and fixed—the assumption can be strictly applied to locate the source of propagation. Such a propagation pattern that strictly obeys the centrality assumption is an indispensable subset that can be covered by the propagation patterns our model can characterize. As the proposed model leverages a simulated label propagation conditional module based on the centrality assumption but employs a graph neural network to learn the influence of the network's heterogeneous topology from the data, other circumstances can also be modeled when learning from the data within our flexible data-driven framework. We have statistically analyzed the closeness centrality (CC) probability density function of the source nodes predicted by our trained ASLDiff model on the *Digg* dataset and compared it with the ground truth centrality of the source nodes in Figure 7(c). The mean and standard deviation for the CCs of the predicted sources are 0.7020 and 0.1444, and that for the CCs of the ground truth sources are 0.7044 and 0.1567, showing that there is no harmful bias in our method's prediction. This statistical result indicates that our model captures the source distribution observed in empirical data, not just theoretical derivations. Our method not only uses knowledge to guide inference to accelerate learning but also learns distribution patterns beyond the knowledge, from the data.

Table 4: Performance under LT diffusion pattern (best with bold).

| Dataset | Jazz | | | | Net | | | | Power | | | |
|---------|------|------|------|------|------|------|------|------|------|------|------|------|
| Method | AC | RE | PR | F1 | AC | RE | PR | F1 | AC | RE | PR | F1 |
| LPSI | 0.985 | **1.000** | 0.777 | 0.875 | 0.900 | **1.000** | 0.322 | 0.487 | 0.752 | **1.000** | 0.168 | 0.288 |
| GCNSI | 0.971 | 0.838 | 0.766 | 0.800 | 0.971 | 0.841 | 0.707 | 0.768 | 0.986 | 0.898 | 0.851 | 0.874 |
| SLVAE | 0.980 | **1.000** | 0.642 | 0.782 | 0.967 | 0.924 | 0.608 | 0.734 | 0.964 | 0.866 | 0.628 | 0.728 |
| ASLDiff(Ours) | **1.000** | **1.000** | **1.000** | **1.000** | **0.978** | 0.828 | **0.782** | **0.804** | **0.990** | 0.889 | **0.899** | **0.952** |

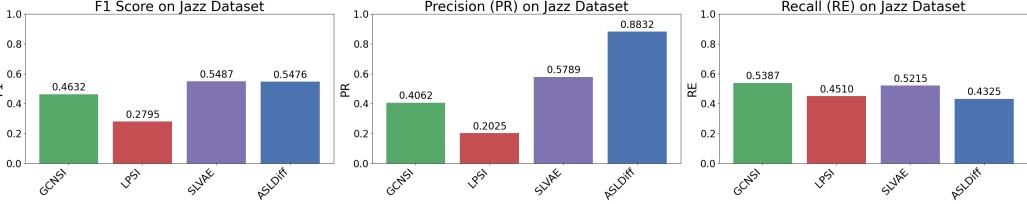

Figure 8: Additional experiments for simulated SIR scenarios on basic performance comparison.

## C  ADDITIONAL RESULTS OF PERFORMANCE UNDER OTHER PROPAGATION PATTERNS

We also test source localization performance under LT and SIR patterns. The experimental results of all synthetic datasets under the LT propagation model are shown in Table 4. The top-performing result for each metric of each dataset has been highlighted in bold for ease of identification. The results show that ASLDiff outperforms all baselines on all datasets under the LT model. Specifically, our method achieves the best performance considering the accuracy (AC), precision (PR), and F1-score (F1) metric, while the recalls are all above 0.8. In the *jazz* dataset, ASLDiff accurately identifies all source nodes and outperforms the second-best method by over 14% in the F1-score, which demonstrates its superiority over the other baselines. Among all the baselines, the non-deep learning method LPSI over-estimates the number of source nodes according to its low precision score, but it still captures all the ground truth sources, indicating its capability to offer valuable advice for a new stage of prediction. ASLDiff takes a step forward over LPSI, hence reaching a better performance.

We conduct additional experiments for simulated SIR scenarios on basic performance comparison. The results are shown in Figure 8. It can be seen that our model can still achieve competitive results compared to these baselines, proving our method's applicability. The results also indicate that in terms of precision, ours achieved the highest score, more than 30% higher than the second-best SLVAE. Although we have a lower recall rate, a decrease of 0.09 only indicates around 1 node is not recalled from the ground truth, as only around 10 nodes are chosen as the ground truth sources in each infection. However, an increase of 0.3 in precision represents around 6 nodes correctly identified without false positives. Therefore, precision should be considered the more critical metric in source localization problems than recall when the F1 scores are similar, and ASLDiff demonstrates the strongest competitiveness among the four methods.

## D  DATASET DESCRIPTION

The detailed description of the adopted datasets is presented as follows.

Table 5: Dataset Overview

| Dataset | Nodes | Edges | Mean Degree | Clustering Coefficient |
|---|---|---|---|---|
| **Jazz** | 198 | 2742 | 13.84 | 0.6174 |
| **Net** | 1589 | 2742 | 1.72 | 0.6377 |
| **Power** | 4941 | 6594 | 1.33 | 0.0801 |
| **Digg** | 14511 | 194405 | 13.39 | 0.1353 |
| **Twitter** | 12619 | 309621 | 24.52 | 0.2962 |

### D.1  SYNTHETIC DATASET

We synthesize propagation data under SIS, IC, and LT models on these three real-world networks: *jazz, network science* and *power grid*. These networks differ in scale, sparsity, and clustering characteristics, which enables us to investigate the model's performance on different types of networks. The statistic overview is presented in Table 5. For the propagation models, the propagation properties of the SIS infection model are determined by the inherent characteristics of the disease, applying homogeneity for all nodes/edges, i.e. the infection and recovery rates in SIS are constant; for the IC and the LT influence model, the heterogeneous propagation probability of each edge is considered, which is set to be inversely proportional to the degree of the target node. This aligns with real-world propagation patterns, where nodes with more connections tend to be less receptive to information from each neighbor.

- *Jazz* (Rossi & Ahmed, 2015). The provided dataset is a network of collaborations among Jazz musicians. Each node in the network represents a musician, and every edge connects two musicians who have performed together in a band. Rumors or infectious diseases are applicable to be propagated on such networks. We randomly choose 5% of nodes to be the

spreading sources of each propagation and use SIS, IC, or LT models to simulate 100 steps or simulate until convergence.

- *Network Science (Net)* (Rossi & Ahmed, 2015). This is a coauthorship network of scientists working on network theory. Nodes represent scientists and edges represent collaborations. Influential information can be propagated on such networks. We randomly choose 0.5% of nodes to be the spreading sources of each propagation and use SIS, IC, or LT models to simulate 100 steps or simulate until convergence.

- *Power Grid (Power)* (Watts & Strogatz, 1998). This is a topology network of the power grid network across the Western United States. In this network, each connection denotes a transmission line for electrical power. The nodes signify one of three components: a power generation unit, a transformer, or a distribution substation. Blackouts can be propagated on such a network. We randomly choose 0.5% of nodes to be the spreading sources of each propagation and use SIS, IC, or LT models to simulate 100 steps or simulate until convergence.

### D.2 REAL-WORLD PROPAGATION DATASET: DIGG

The datasets selected, Digg and Twitter, represent real-world social networks where information propagation can be authentically traced, and are commonly used for evaluation in previous works Ling et al. (2022); Huang et al. (2023a). Both include propagation cascades demonstrating the time stamps and the information diffusion trace among users of each post or message. A connection network of all users is also provided in each dataset. Both datasets are pertinent to our study because they exemplify real-world dynamics of information spread.

Digg (Rossi & Ahmed, 2015) is real-world social network data showcasing voting records of stories that made it to Digg's front page in 2009, with each story's spread counted as one diffusion cascade. We randomly choose 100 stories to form our dataset. The nodes (voters) involved in these stories form a subgraph of the original one, where the links represent the friendship of voters. The statistics of this friendship network are shown in Table 5. Drawing an analogy to the spread of a virus during a pandemic, it is often difficult to detect the virus at the very beginning, but after some time has passed—such as when the manifestation of symptoms—we can observe the infection status of the population. As a result, for each story cascade, we choose the top 10% of nodes and 30% of nodes as diffusion sources and observed influenced nodes based on their influenced time.

In section 5.3, we also perform simulations on Digg of the SIS and the IC model for few-shot experiments. In the pretrain dataset preparation, we hold the network's topology and randomly choose between 0.15% and 1.5% of nodes to be the spreading sources of each propagation. We then use the SIS and the IC model to simulate 100 steps or until convergence.

### D.3 REAL-WORLD PROPAGATION DATASET: TWITTER

The Twitter (Yang et al., 2021) dataset is a collection of social network and public tweets written in English that were posted on the social media platform Twitter (a.k.a X) from March 24th to April 25th, 2012. The network statistics are shown in Table 5. Each tweet can be counted as one propagation cascade. Same as *Digg*, for each cascade, we choose the top 10% of nodes and 30% of nodes as diffusion sources and observed influenced nodes based on their influenced time.

In section 5.3, we also perform simulations on Twitter of the SIS and the IC model for few-shot experiments. In the pretrain dataset preparation, we hold the network's topology and randomly choose between 0.15% and 1.5% of nodes to be the spreading sources of each propagation. We then use the SIS and the IC model to simulate 100 steps or until convergence.

### D.4 SIMULATED NETWORK DATASETS USED IN SECTION 5.3.2

We employ several established network generation algorithms to create multiple random networks: Erdős-Rényi (ERDdS & R&wi, 1959), Barabási–Albert (Barabási & Albert, 1999), and Watts-Strogatz (Watts & Strogatz, 1998). These networks vary in size, with node counts ranging from 1,000 to 10,000. We present the parameters and statistics of the simulated dataset of each random network in Table 6. For each generated network, we simulate the SIS (Susceptible-Infected-

| Network Model | Count | Model Parameters |
|---|---|---|
| Erdős-Rényi (ER) | 500 | $n$ (number of nodes): 200-1000
$p$ (connection probability): 0.0020-0.0030 |
| Barabási-Albert (BA) | 500 | $n$ (number of nodes): 200-1000
$m$ (edges added per new node): $(0.010 - 0.015)n$ |
| Watts-Strogatz (WS) | 500 | $n$ (number of nodes): 200-1000
$K$ (initial neighbors per node): $(0.010 - 0.015)n$
$p$ (rewiring probability): 0.4 |

Table 6: Network Models and Their Parameters

Susceptible) and IC (Independent Cascade) models. The results are then organized into a synthetic dataset, which is categorized by both the propagation pattern and the network generation model. They are then used for generalization experiments in Section 5.3.2.

## E  BASELINES

We compare the performance of ASLDiff against three state-of-the-art baselines of source localization methods using propagation snapshot observations. To the best of our knowledge, these methods are the only ones that illustrate their superiority against other works on locating sources without knowing the underlying propagation pattern, which is the same as ours. The detailed information is presented as follows.

- **NetSleuth** (Prakash et al., 2012) utilizes a minimum description length approach to filter nodes from multiple sources, yet it is exclusively designed to operate within the Susceptible-Infected (SI) model framework.

- **LPSI** (Wang et al., 2017) is a novel method for detecting multiple sources of information diffusion in networks without a predefined propagation model, leveraging the concept of source prominence and label propagation to identify probable sources based on local peaks in the propagation landscape. In our experiment, the parameter $\alpha$ in LPSI is determined by testing it among the values {0.1, 0.3, 0.5, 0.7, 0.9} for each evaluation dataset and then selecting the best one.

- **GCNSI** (Dong et al., 2019) introduces a deep learning approach for identifying multiple rumor sources in social networks without needing the underlying propagation model, using graph convolutional networks to enhance prediction precision through spectral domain convolution and multi-order neighbor information. The setting of this model follows the description in (Dong et al., 2019).

- **SLVAE** (Ling et al., 2022) is a probabilistic framework designed to tackle the challenge of source localization in graph diffusion problems using a variational autoencoder approach to quantify uncertainty and leverage prior knowledge. We follow the original implementation in the paper, tune the learning rate from 0.001 to 0.05, and select the best one.

- **TGASI** (Hou et al., 2023) is a sequence-to-sequence framework for multiple rumor source detection that considers heterogeneous user behavior in time-varying scenarios. It uses a GNN-based encoder to generate multiple features and a GRU-based decoder with temporal attention to infer sources. TGASI is designed with transferability and uses a unique loss function.

- **DDMSL** (Yan et al., 2024) proposes a novel probabilistic model for source localization and diffusion path reconstruction in complex networks. By formulating information propagation as a discrete diffusion process, DDMSL employs a reversible residual network to construct a denoising-diffusion model in discrete space. This approach allows for both accurate source identification and comprehensive reconstruction of information diffusion pathways.

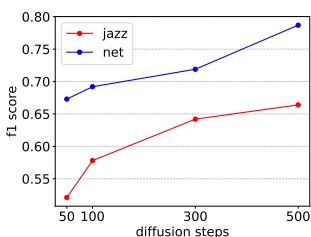 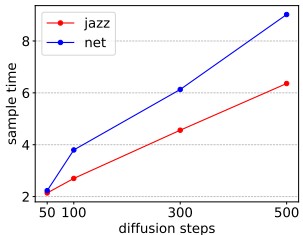

Figure 9: F1 score vs. diffusion step under SIS.

Figure 10: Sample time vs. diffusion step under SIS.

## F    EXPERIMENTS AND IMPLEMENTATION DETAILS

For each dataset, the ratio of training, validation, and testing portion is 6:1:1. For the diffusion framework of ASLDiff, we use $T = 500$ maximum diffusion timestep and linear schedule for noise scheduling. In the denoising network, we leverage a 2-layer graph convolutional network (GCN) to forward the LPSI estimation $X_{est}$. The GNN encoder and decoder comprise 3-layer GCNs with a hidden dimension of 128. The residual GNN of the conditioning module is a 2-layer GCN, with a hidden dimension of 8. The learning rate is searched from 0.01, 0.005, 0.001, and the maximum number of training epochs is set to 500 for all datasets. In the few-shot learning experiments, the maximum pretrain/finetune epoch is set to 300. We train our model using Adam optimizer and a learning rate scheduler with a linear decay. Our model is trained on a single NVIDIA GeForce RTX 2080 Ti. The code implementation can be found at https://anonymous.4open.science/r/ASLDiff-4FE0.

## G    PARAMETER ANALYSIS: DIFFUSION STEP

We perform additional experiments on how the maximum diffusion step affects the performance and time consumption of ASLDiff on *jazz* and *network science* datasets, which use the SIS propagation model for data synthesis. The results are shown in Figure 9 and Figure  10. It can be observed that when the diffusion step increases, the performance also improves. Specifically, the improvement from the lowest to the highest f1-score in the *jazz* dataset is about 27%, which is higher than that in the *network science* dataset (17%). We also evaluate the performance when the diffusion step becomes 1, which makes the model a VAE, and ASLDiff (F1) performance drops by nearly 90%. The performance and diffusion steps show a positive correlation, demonstrating the beneficial effect of the diffusion framework. Sampling difficulty decreases as noise is more accurately added in the forward diffusion process. It is also reasonable that the sampling time increases as the diffusion step becomes higher. Hence, the tradeoff should be clearly considered when choosing the appropriate maximum diffusion timestep.

## H    LIMITATIONS

Our proposed method also exhibits certain limitations. Our approach may, to some extent, depend on the accuracy of the advice provided by soft labels, despite our application of various sophisticated designs to enhance the model's adaptability. As a result, when confronted with more complex scenarios, our method might reveal limitations. On the other hand, the sampling speed of our multi-step diffusion model may be slower compared to some deep learning methods, which could become a bottleneck for applications requiring real-time localization. While computational constraints currently limit our model's direct application to million-node networks, the core principles we developed can be integrated into hierarchical approaches. This hierarchical strategy would effectively reduce the network scale, allowing us to leverage our method's proven strength in accurate source localization for moderately-sized networks. We will continue to conduct in-depth research in these areas.

Table 7: Model Time Performance Analysis

| Method | Training Time (h) |
|---|---|
| SLVAE | $\sim$2.5 |
| GCNSI | $\sim$1.5 |
| TGASI | $\sim$2.5 |
| DDMSL | $\sim$3 |
| ASLDiff | $\sim$3 |
| ASLDiff pretrain+few-shot | $\sim$1 + $\sim$0.5 |

*(a) Comparison of average training time (in minutes) for different methods.*

| Method | Inference Time (s) |
|---|---|
| LPSI | 0.167 |
| SLVAE | 0.500 |
| GCNSI | 0.333 |
| TGASI | 0.333 |
| DDMSL | 0.500 |
| ASLDiff | 0.667 |

*(b) Average inference time per sample (in seconds) for different methods.*

## I COMPUTATIONAL COMPLEXITY ANALYSIS

We present a detailed comparison of the computational cost of our proposed model against baselines, as outlined in Table 7.

In terms of computational efficiency, ASLDiff achieves a reasonable training time, requiring less than 4 hours on a single RTX-2080 Ti GPU. It is notable that:

(1) While our model's training duration is marginally higher due to the iterative nature of the DDPM-based denoising process, this initial time investment can be offset by ASLDiff's crucial advantage: its few-shot and zero-shot adaptability to various networks and patterns, which significantly reduces computational resources in real-world applications. As shown in the table, our model requires less time to pretrain on the synthetic network and finetune on the target network(correspond to Section 5.3.2) than initially training on the target network since the scale of the network in the pretrain data can be smaller.

(2) We opted for DDPM as our foundation due to its classical design and proven effectiveness. It's worth highlighting that ASLDiff's architecture is fully compatible with more computationally efficient diffusion variants, such as DDIM [1], which could substantially reduce the current computational overhead. This flexibility, combined with our model's transfer capabilities, makes ASLDiff particularly resource-efficient in practical deployments.

