# OpenReview forum: "Adaptive Source Localization on Complex Networks via Conditional Diffusion Model"
_ICLR.cc/2025/Conference — Submitted to ICLR 2025_

### Official Review · Reviewer_U3Da · 2024-10-24

**Soundness:** 3
**Presentation:** 3
**Contribution:** 2
**Rating:** 5
**Confidence:** 3

**Summary:**

The paper studies the source localization problem: given a graph with infection states of all nodes, aim to find the original propagation source. The paper proposes ASLDiff which utilizes diffusion model for source localization in complex networks and applies GNNs to enhance the model’s adaptability to diverse network topologies. Besides, it incorporates soft labels and a restructured label propagation process to capture essential propagation characteristics across various network topologies. However, the technique novelty is limited and methods are capped which subject to labelling dissemination methods.

**Strengths:**

S1. The problem of source localization is important.

S2. The problem statement and the rationale behind the method are clear.

S3. The paper is a reasonable attempt to apply diffusion modelling to traditional propagation problems.

S4. The evaluation is very comprehensive. The authors show strong quantitative results. Tables 1-2, with figs 3-6 give strong qualitative results and ablation.

**Weaknesses:**

W1. The forward process (e.g. the diffusion process) should be detailed. What is the state of the initial $X$ and what is the $X$ after noise added? Besides, it seems that you used the conditional diffusion method, but your condition is an approximate solution obtained from LPSI. This usually does not conform to our intuition, and I hope you can elaborate on the rationality of using the approximate solution of LPSI as a condition. And will using approximate solutions as conditions greatly limit the performance of your proposed model? If this impact does not occur, you should provide some theoretical proof. This is evident in the accuracy results, where ASLDiff underperforms compared to LPSI on some metrics.

W2. The description of the method in the paper is unclear, and the innovative aspects of the technical details are not sufficiently articulated. For example, the part of the LPSI advisor that employs the conditional diffusion model lacks an explanation of how the diffusion model introduces innovation for the specific task.

W3. The framework diagrams in the paper are not sufficiently clear. For instance, in Figure 1, the function $f(\theta)$, which corresponds to the denoising network, is not intuitive. It is suggested to use clearer module diagrams to explain this. Additionally, the input-output relationships in Figure 2 are unclear. For example, the concept of $Y_{label}$ is not explicitly defined in the text, even though the generation method is explained. Further clarification is recommended.

**Questions:**

If LPSI is used to generate enough training samples to train the DL-based method, is it possible to achieve an approximation with adaptability, and what are the advantages of using a diffusion model in comparison?

---

> ### Author Response · Authors · 2024-11-20
>
> Thanks for your constructive feedback. We have summarized your comments into the following questions. We hope the following responses can address your concerns.
>
> **1. Detailing the forward process**
>
> We appreciate the reviewer's questions about our diffusion process and the use of LPSI estimates as conditions. The initial state $X_0$ is a binary vector indicating source nodes (1 for source, 0 otherwise). Through the diffusion process defined by equations (4) and (5), noise is gradually added such that $X_t$ follows a Gaussian distribution with mean determined by both the original X0 and the LPSI estimate $X_{est}$. The process eventually converges to Xn following $N(X_{est}, I)$, providing a smooth transition from binary labels to continuous distributions.
>
> **2. Problems of using the approximate solution of LPSI**
>
> Using approximate solutions as conditions does not greatly limit the performance of the proposed model. We don't treat LPSI results as hard constraints, but rather as a prior guide- the ending point of diffusion. The denoising process allows the diffusion model to explore broader solution space beyond LPSI's local optima.
>
> The ablation studies in Section 5.4 show that our diffusion model without prior-guided design obtains a performance drop of 1.7~17%, substantiating that this design effectively enhances model performance without being limited by LPSI's approximations. Also, in the main experiment, while LPSI occasionally shows higher recall on Jazz and Power datasets, this is largely due to its tendency to over-predict source nodes, as evidenced by its consistently lower precision(0.083/0.619/0.775/0.498 compared to ours 0.635/0.902/0.849/0.507(-87%/-31%/-9%/-2%)) scores in Table 1&2. More importantly, ASLDiff achieves superior overall F1 scores across datasets, demonstrating that it successfully leverages but is not limited by LPSI's predictions.
>
> [1] Devavrat Shah and Tauhid Zaman. Rumors in a network: Who’s the culprit? IEEE Transactions on Information Theory, 57(8):5163–5181, 2011.
>
> **3. The innovative aspects of the technical details are unclear**
>
> We thank the reviewer for highlighting the need for a clearer articulation of our method's technical innovations. Our diffusion model introduces specific innovations that directly address two fundamental challenges in source localization.
> Firstly, source localization suffers from high uncertainty and limited training data. To address this, we innovatively modify the diffusion framework's endpoint distribution to incorporate domain knowledge about source characteristics. Specifically, instead of using standard Gaussian noise as the endpoint, we align it with LPSI-derived source estimates that encode well-established principles of source prominence and centrality. This allows our model to leverage reliable prior knowledge while maintaining the flexibility to explore the full solution space through the denoising process. The effectiveness of this innovation is demonstrated in our ablation studies, where removing this prior-guided diffusion leads to a significant performance drop(1.7~17%) across all datasets.
>
> Secondly, source localization requires understanding complex, heterogeneous propagation patterns across different network structures. We address this through an innovative GCN-based parameterization of the label propagation process in our denoising network, unlike standard diffusion models that use generic neural architectures. The GCN structure allows the model to learn adaptive propagation rules by combining fixed theoretical principles (encoded in LPSI's label propagation) with data-driven features (similar to GCNSI). The effectiveness of this innovation is demonstrated in our ablation studies, where removing this propagation-enhanced conditioning module leads to a significant performance drop (0.5~40%) across all datasets.
>
> To improve clarity, we have enhanced our manuscript with more detailed explanations of these innovations, particularly in Sections 4.1 and 4.2. We now provide more explicit descriptions of how our GCN-based parameterization enables adaptive learning.
>
> **4. Revision of the framework diagrams**
>
> We have improved the clarity of our framework diagrams in the revised manuscript. For Figure 1, we have replaced $f_\theta$ with 'denoiser,' which is used more commonly. For Figure 2, we have enhanced the visualization with clear input and output and expanded Section 4.2 to provide more explicit descriptions of the architecture components and their relationships.

---

> > ### Author Response · Authors · 2024-11-20
> >
> > **5. Advantages of using a diffusion model compared with LPSI-trained DL method**
> >
> > Training deep learning models purely on LPSI-generated samples would constrain the learned distribution to LPSI's inherent limitations and biases, potentially leading to local optima that mirror LPSI's deterministic nature. This would significantly limit the model's ability to adapt to novel propagation patterns.
> > Our diffusion model approach instead leverages LPSI outputs as soft prior guides rather than hard constraints. By using LPSI estimates to guide the diffusion endpoints while maintaining flexibility in the denoising process, we achieve a balanced combination of knowledge-driven and data-driven learning. This allows our model to capture patterns beyond LPSI's assumptions while benefiting from its proven effectiveness.
> > The diffusion framework also provides two key advantages: First, it excels at capturing the intrinsic uncertainty in source locations. Second, it enables elegant integration of prior knowledge through our soft-label guided approach, helping mitigate the impact of limited training data.
> > The effectiveness of the diffusion model is evidenced by experiments in Appendix G, where reducing diffusion steps from 500 to 1 leads to a ~90% drop in the F1 score on the jazz dataset.

---

> > > ### Comment · Reviewer_U3Da · 2024-11-22
> > >
> > > I appreciate the author's response and I have read all the reviewers' comments. I tend to keep the current score.

---

> > > > ### Author Response · Authors · 2024-12-03
> > > >
> > > > Dear reviewer,
> > > >
> > > > We have provided supplementary extensive experiments on two representative networks with distinctly different characteristics to demonstrate the effectiveness of ASLDiff on SIS (comment to reviewer h6eH) and demonstrated our work's highlight in the global comment. We sincerely hope you reconsider the score and provide further constructive feedback in light of these additional experiments and clarifications.
> > > >
> > > > Thank you for your time and consideration.

---

### Official Review · Reviewer_h6eH · 2024-10-30

**Soundness:** 2
**Presentation:** 3
**Contribution:** 2
**Rating:** 5
**Confidence:** 5

**Summary:**

This paper proposes an Adaptive Source Localization Diffusion Model to face the challenge of data scarcity without specific propagation models. And evaluations of various propagation patterns and real network datasets demonstrate ASLDiff’s effectiveness.

**Strengths:**

1. This paper provides comprehensive experimental validation of ASLDiff.
2. The code of ASLDiff is given.
3. ASLDiff shows significant improvement on some datasets.

**Weaknesses:**

1. As the authors mention, the diffusion model is highly complex. However, the extent of this complexity is not quantified. Additionally, considering the high complexity, the motivation for using the diffusion model should be fully explained.
2. The authors discuss the challenge that "real-world networks typically exhibit unknown propagation patterns", but they do not explain or demonstrate how ASLDiff understands different propagation patterns in different scenarios.
3. Sections 2.1 and 3.2 are confusing, as they both seem to explain the IC and LT models redundantly.
4. The authors have demonstrated that the closeness centrality of infected nodes and sources is very consistent, but it appears that the advisor used in ASLDiff is not based on closeness centrality.

**Questions:**

see Weaknesses

---

> ### Author Response · Authors · 2024-11-20
>
> Thanks for your constructive feedback. We have summarized your comments into the following questions. We hope the following responses can address your concerns.
>
> **1. Quantification of ASLDiff's complexity**
>
> We acknowledge the reviewer's concern about the complexity of our diffusion-based approach. In response, we present a detailed comparison of the computational cost of our proposed model against baselines on the Twitter Dataset, as outlined in the table below:
>
> **Average training time on Twitter Dataset**
>
> | Metrics  | SLVAE | GCNSI | TGASI | DDMSL | ASLDiff | ASLDiff's pretraining + few-shot|
> |----------|-------|-------|--------|--------|----------|---|
> | Avg Training Time  | ~2.5h | ~1.5h | ~2.5h | ~3h | ~3h | ~1h + ~0.5h|
>
> **Average inference time per sample**
>
> | Method | LPSI | SLVAE | GCNSI | TGASI | DDMSL | ASLDiff |
> |----------|-------|-------|-------|--------|--------|----------|
> | Avg Inference Time | ~10s | ~30s | ~20s | ~20s | ~30s | ~40s |
>
> In terms of computational efficiency, ASLDiff achieves a reasonable training time, requiring less than 4 hours on a single RTX-2080 Ti GPU.
> It is notable that:
> - While our model's training duration is marginally higher due to the iterative nature of the DDPM-based denoising process, **this initial time investment can be offset by ASLDiff's crucial advantage: its few-shot and zero-shot adaptability to various networks and patterns, which significantly reduces computational resources in real-world applications.** As shown in the table, our model requires less time to pretrain on the synthetic network and finetune on the target network(correspond to Section 5.3.2) than initially training on the target network since the scale of the network in the pretrain data can be smaller.
> - We opted for DDPM as our foundation due to its classical design and proven effectiveness. It's worth highlighting that ASLDiff's architecture is fully compatible with more computationally efficient diffusion variants, such as DDIM [1], which could substantially reduce the current computational overhead. This flexibility, combined with our model's transfer capabilities, makes ASLDiff particularly resource-efficient in practical deployments. We have incorporated a detailed efficiency analysis in Appendix I of the revised manuscript.
>
> [1] Song, Jiaming, Chenlin Meng, and Stefano Ermon. "Denoising Diffusion Implicit Models." International Conference on Learning Representations. 2020.
>
> **2. The motivation for using the diffusion model**
>
> While ASLDiff does indeed have higher computational requirements, our approach leverages diffusion models for their strong capability in modeling the uncertainty of sources and incorporating domain knowledge. The advantages are two-fold: First, as the current state-of-the-art probabilistic generative framework, the diffusion model is able to capture the intrinsic uncertainty in source locations - understanding "why certain locations are more likely to be sources" by learning the underlying probability distributions. Second, the iterative diffusion process enables fine-grained integration of prior knowledge at multiple scales through its progressive denoising steps. The diffusion framework allows for elegant integration of prior knowledge through our innovative soft-label guided approach, which helps mitigate the impact of limited training data.
>
> The effectiveness of the diffusion model is evidenced by experiments in Appendix G, where reducing diffusion steps from 500 to 1 leads to a ~90% drop in the F1 score on the jazz dataset. When diffusion steps=1, the framework generally reduces to a VAE. The result demonstrates that the multi-step process helps incorporate domain knowledge more effectively than VAE.

---

> ### Author Response · Authors · 2024-11-20
>
> **3. Demonstration on how ASLDiff understands different propagation patterns**
>
> We appreciate the reviewer's comment about explaining how ASLDiff understands different propagation patterns.  To understand different patterns, unlike existing methods that rely purely on data to gain an understanding of the propagation patterns, our method introduces guidance from **the universal properties** across patterns--**source centrality and prominence** that are evidenced through empirical analysis (Appendix B) and existing research [1]. ASLDiff introduces these through label propagation-based prior that encodes source prominence and centrality principles, which allows ASLDiff to pretrain with synthetic data for learning universal propagation knowledge while maintaining the flexibility to quickly adapt to unknown real-world patterns with minimal data requirements. As evidence, the experiment shown in Figure 4 shows that pretrained ASLDiff requires only 3% fine-tuning data on the Digg dataset to achieve optimal performance.
>
> Furthermore, **additional experiments** are conducted here to better investigate how ASLDiff models different patterns. We extend the experiment in Section 5.3.1, adjust the pattern categories of synthetic pretrain data and remove the prior guidance. The result is shown below:
>
> | Pretraining Setup | Zero-shot Performance (F1) |Few-shot(15%) Performance (F1)|
> |----------------|---------------------------|---|
> |Without pretrain|-|0.0585|
> | Only IC | 0.3298 |0.4149|
> | Only SIS | 0.0349 |0.3303|
> | IC+SIS w/o Prior-guided Diffusion | 0.4974 |0.5560|
> | IC+SIS (Full Model) | 0.5574 |0.5826|
>
> From the result, we can find that:
> 1. Combined IC+SIS pretraining (F1=0.5574) significantly outperforms single-pattern training (IC: 0.3298, SIS: 0.0349), demonstrating enhanced pretraining from **diverse pattern exposure**.
> 2. Zero-shot results with IC-only pretraining substantially exceed SIS-only pretraining（0.3298 v.s. 0.0349）, indicating that Twitter's real-world propagation aligns closer with IC patterns, consistent with IC's design for information diffusion modeling.
> 3. Fine-tuning with domain data improves performance over no pretraining (0.4149/0.3303 v.s. 0.0585), validating **the value of pretrained pattern knowledge**.
> 4. Removing prior-guided diffusion reduces performance (0.4974 vs 0.5574), confirming the **importance of universal property guidance.**
>
> To conclude, the superior pattern understanding capability of ASLDiff stems from:
>
> - **Rich synthetic pretraining data incorporating diverse propagation patterns** (IC+SIS)
> - **Universal property guidance through prior-guided diffusion**, which captures pattern-invariant properties while maintaining adaptability to pattern-specific features
>
> [1] Devavrat Shah and Tauhid Zaman. Rumors in a network: Who’s the culprit? IEEE Transactions on Information Theory, 57(8):5163–5181, 2011.
>
> **4. Conflicts between Section 2.1 and Section 3.2**
>
> Thank you for this insightful comment. We agree that the current organization of Sections 2.1 and 3.2 creates unnecessary redundancy in explaining the IC and LT models. We have merged these two sections into one as Section 3.2 to reduce redundancy. The revision can be checked in the highlighted red contents.
>
> **5. The advisor used in ASLDiff**
>
> The advisor used in ASLDiff can capture the centrality of infected nodes by calculating soft-labels using the converged form of equation 1. In each iteration using equation 1, the label of a node is updated based on its neighbors’ labels. The initial labels(1 or -1) indicate whether a node is infected or not. In the end, the node with higher centrality will be influenced more by the infected nodes, hence obtaining different labels compared to those with less centrality. Therefore, the output of the advisor encodes the feature of centrality, which is a meaningful indicator of ground truth sources.

---

> ### Author Response · Authors · 2024-11-22
>
> We appreciate your invaluable time and insightful comments. We have provided more details to answer your questions, clarified our motivation, and added experiments to demonstrate how ASLDiff understands different propagation patterns.
>
> We have also renewed our manuscript (https://openreview.net/pdf?id=v0O9FrVTt1) to reduce the redundancy in the original paper.
>
> Can you kindly check them and let us know if they address your concerns? We are happy to answer any further questions you may have.

---

> > ### Comment · Reviewer_h6eH · 2024-11-26
> >
> > I appreciate the author's response and have read all the reviewer's comments. Similarly, I am also concerned about the poor performance of the model on SIS, but the author has not provided a convincing answer. Additionally, ASLDiff does not have an advantage in terms of inference time. Based on this, I tend to keep the current score.

---

> > > ### Author Response · Authors · 2024-11-27
> > >
> > > Thanks for your feedback. We conducted extensive experiments on two representative networks with distinctly different characteristics to demonstrate the effectiveness of ASLDiff on SIS:
> > >
> > > **Table 1: Results on Digg Network with SIS Model**
> > >
> > > Digg (Rossi & Ahmed, 2015), adopted in SLVAE, is a real-world social network dataset showing voting records of stories that made it to Digg's front page in 2009. The network contains 14,511 nodes and 194,405 edges. The experiment uses the SIS propagation model with parameters:
> > >
> > > - Distribution of initial seed ratio: Uniform(0.0015, 0.015)
> > > - Infection probability: 0.0001
> > > - Recovery probability: 0.00005
> > >
> > > | Model   | F1    | RE    | PR    | AC    |
> > > |---------|-------|-------|-------|-------|
> > > | LPSI    | 0.8265| **0.8969**| 0.769 | 0.9968|
> > > | GCNSI   | 0.5619| 0.5043| 0.6344| 0.9643|
> > > | DDMSL   | 0.7586| 0.722 | 0.8001| 0.9832|
> > > | Ours    | **0.8403**| 0.8307| **0.852**| **0.9973**|
> > >
> > > **Table 2: Results on ego-Facebook Network with SIS Model**
> > >
> > > Ego-Facebook(J. McAuley & J. Leskovec, 2012), adopted in GCNSI, contains social circles from Facebook, consisting of 4,039 nodes and 88,233 edges. The experiment uses the SIS propagation model with parameters:
> > >
> > > - Distribution of initial seed ratio: Uniform(0.001, 0.003)
> > > - Infection probability: 0.0001
> > > - Recovery probability: 0.00002
> > >
> > > | Model   | F1    | RE    | PR    | AC    |
> > > |---------|-------|-------|-------|-------|
> > > | LPSI    | 0.7757| 0.8365| 0.7545| 0.9990|
> > > | GCNSI   | 0.4720| 0.4086| 0.5588| 0.9976|
> > > | DDMSL   | 0.5659| 0.5379| 0.6196| 0.9977|
> > > | Ours    | **0.8316**| **0.8570**| **0.8189**| **0.9992**|
> > >
> > > These results demonstrate strong consistency in that ASLDiff maintains superior F1, PR, and AC performance across networks with different scales and characteristics. Together with the results in the manuscript, these comprehensive evaluations strongly support ASLDiff's effectiveness under the SIS diffusion pattern.

---

> > > ### Author Response · Authors · 2024-12-03
> > >
> > > In terms of inference time, we have conducted extensive experiments using efficient diffusion variants DDIM to reduce the denoising steps from 500 to 250, and we have observed slight changes in performance, as shown in the following table:
> > >
> > > | Dataset    | Method                    | F1-score | Inference Time (s) |
> > > |------------|---------------------------|----------|---------------------|
> > > | **Jazz SIS**   | Ours using DDPM 500 steps  | 0.720    | ~40s                |
> > > |            | Ours using DDIM 250 steps  | 0.696    | ~25s                |
> > > | **Jazz IC**   | Ours using DDPM 500 steps  | 0.901    | ~40s                |
> > > |            | Ours using DDIM 250 steps  | 0.882    | ~20s                |
> > > | **Net SIS**    | Ours using DDPM 500 steps  | 0.816    | ~50s                |
> > > |            | Ours using DDIM 250 steps  | 0.801    | ~20s                |
> > > | **Net IC**   | Ours using DDPM 500 steps  | 0.480    | ~50s                |
> > > |            | Ours using DDIM 250 steps  | 0.475    | ~25s                |
> > > | **Power SIS**  | Ours using DDPM 500 steps  | 0.877    | ~50s                |
> > > |            | Ours using DDIM 250 steps  | 0.876    | ~25s                |
> > > | **Power IC**   | Ours using DDPM 500 steps  | 0.516    | ~50s                |
> > > |            | Ours using DDIM 250 steps  | 0.517    | ~20s                |
> > >
> > > Our model's performance does not decrease too much among all datasets(at most -3.3%), but the inference times are significantly reduced to nearly 50%, after using the DDIM with reduced steps. The results show that our model can reduce the current computational overhead while keeping the main performance with the help of the computationally efficient diffusion variant. This flexibility, combined with our model's transfer capabilities, makes ASLDiff particularly resource-efficient in practical deployments.

---

> ### Author Response · Authors · 2024-12-03
>
> Dear reviewer,
>
> We have provided supplementary extensive experiments on two representative networks with distinctly different characteristics to demonstrate the effectiveness of ASLDiff on SIS and demonstrated our work's highlight in the global comment. We sincerely hope you reconsider the score and provide further constructive feedback in light of these additional experiments and clarifications.
>
> Thank you for your time and consideration.

---

### Official Review · Reviewer_NsyM · 2024-10-31

**Soundness:** 2
**Presentation:** 4
**Contribution:** 3
**Rating:** 5
**Confidence:** 4

**Summary:**

In the social network background, the author employs the diffusion model to implement source detection tasks across different localization scenarios. The proposed ASLDiff can be trained in a few-shot manner.

**Strengths:**

ASLDiff can train based on the few-shot learning.

The appendix contains a sufficient review of related work, indicating the adequate preparations for the proposed ASLDiff.

ASLDiff using a single snapshot is superior to multiple-snapshot-based works. I think it is awesome!

**Weaknesses:**

One of the contributions of this paper is the development of few-shot learning due to the limited data available in real-world scenarios. This should be a key focus for the author; however, it seems that the main body of the paper provides a very limited description of few-shot learning. Moreover, despite the thorough survey and preparations of the work, there are many details that need attention. For example, 'Diffusion' instead of 'Diffsion';  Equation (1),  Equation (7) instead of Equation 1,  equation 7. Also, providing code is intended to enhance confidence for your work, but the code is not runnable.

**Questions:**

The authors’ research scope is so wide that it may be incredible. They claim that they can solve problems in various fields, such as disease outbreaks, network security, etc. This is a great plan, however, maybe it is an exaggeration. The spread of infectious diseases and the spread of misinformation vary greatly. Maybe it is too ideal to solve these problems with only one model? This is my main concern for this paper, as complex networks are a very vast and intricate field.

The author's review is complete, but there are relatively few comparative methods published in the year 2024. Compared with the newest algorithms in the year 2024, can ASLDiff still maintain its advantage?

After setting up the required environment for the author's code, I encountered an error stating "models.guide: No such file or directory." I can't verify the results of the model, even though it is impressive.

---

> ### Author Response · Authors · 2024-11-20
>
> Thanks for your constructive feedback. We have summarized your comments into the following questions. We hope the following responses can address your concerns.
>
> **1. About our research scope**
>
> Thank you for raising this important concern. We would like to emphasize that we do not want to propose a universal model that handles the source localization problem in different propagation scenarios that have distinct underlying mechanisms. In applications, our proposed model is supposed to be trained individually in different scenarios. We have revised our misleading statement that may drive the readers to consider our model to be a universal model in our renewed manuscript in red.
>
> The core of our proposed method is introducing the knowledge that is universal across propagation patterns: source prominence and centrality. The former comes from the common observation that sources are surrounded by more infected nodes, while the centrality of sources shows that nodes far from the source are less likely to be infected than those near it. The centrality of sources is proved to be observed in the real-world data by our analysis in Appendix B. In our analysis of the real-world dataset Digg, the mean normalized closeness centrality of sources is higher than the average of all infected nodes, and source nodes cover over 63%
> of the nodes with the centrality score exceeding 0.8, overall demonstrating the sources' higher likelihood of being central to the network structure within the cascades.
>
> By introducing the knowledge into our diffusion framework, two benefits can be obtained: (1) when sufficient domain data is available, it can help the model capture characteristics of propagation pattern more effectively, since in Figure 4, our non-pretrained model performs consistently better than the learning-based GCNSI in both Digg and Twitter. (2) When domain data is limited, our experiments in Section 5.3.1 demonstrate that the model can be pretrained on synthetic propagation data simulated on established propagation models and obtain relatively good few-shot or zero-shot performance. This is because our model can effectively learn pattern-invariant features from pretrain data under the enhancement of knowledge, which is more practical in real-world cases. However, we are still not proposing a singular trained model to tackle multiple scenarios. Training a foundation model based on synthetic and multi-scenario real-world data is an ambitious and challenging problem that is out of our scope.
>
> **2. The limited description of few-shot learning**
>
> We agree that the few-shot learning capability is a key contribution that deserves more thorough elaboration. We have added detailed discussions on the model's potential few-shot learning capabilities in the method session, while the results section provided comprehensive descriptions of the experimental setup and results analysis. The revised part is highlighted in red in the manuscript.
>
> **3. Formatting and typographical issues**
>
> Thank you for catching these inconsistencies. We have conducted a thorough editorial review to ensure consistent formatting and terminology. The revised part is highlighted in red in the manuscript.
>
> **4. Code implementation**
>
> We apologize for any issues with the code execution. We have made the repository runnable and provided clear documentation and setup instructions. Please check the original link: https://anonymous.4open.science/r/ASLDiff-4FE0.
>
> **5. Discussion on methods from 2024**
>
> As shown in Section 5, we have conducted comparisons with DDMSL (Yan et al., 2024), one of the newest state-of-the-art approaches from NeuIPS 2023(December), and our results show that ASLDiff consistently achieves improvements in F1 scores across different datasets. There are a few other methods published in early 2024, like PGSL[1] and GINSD[2]. PGSL resembles SLVAE's framework and merely utilizes a flow-based model to replace the VAE in SLVAE, while our diffusion model exhibits stronger distribution modeling capabilities. GINSD considers incomplete user data scenarios and utilizes a positional embedding module to distinguish incomplete nodes in the source inference process, and as we do not consider such circumstances, GINSD reduces to a simple GAT-based baseline similar to GCNSI. What's more, neither method has been open-sourced for direct empirical comparison. We have added a comparison of the two methods in Appendix A.
>
> [1]Xu, Xovee, et al. "PGSL: A probabilistic graph diffusion model for source localization." Expert Systems with Applications 238 (2024): 122028.
> [2]Cheng, Le, et al. "GIN-SD: source detection in graphs with incomplete nodes via positional encoding and attentive fusion." Proceedings of the AAAI Conference on Artificial Intelligence. Vol. 38. No. 1. 2024.

---

> > ### Comment · Reviewer_NsyM · 2024-11-22
> >
> > The author states that the revised part is highlighted in red in the manuscript. However, I could not find the new manuscript, so I cannot assess the author's rebuttal. Based on the above considerations, I am keeping the score unchanged.

---

> ### Author Response · Authors · 2024-11-22
> **Regarding the location of the new manuscript**
>
> Apologies for not specifying the location of the new manuscript. You can find it by clicking the PDF link located on the top right side of this webpage, or via https://openreview.net/pdf?id=v0O9FrVTt1.

---

> > ### Author Response · Authors · 2024-12-03
> >
> > Dear reviewer,
> >
> > We appreciate your valuable time and high-quality review. This is a kind reminder that the dicussion phase will be ending soon on December 2th 11:59pm AOE. We hope our answers and extensive experiments have addressed your concerns and questions. Please let us know if you have any more questions before the end of the discussion period.
> >
> > Thank you for your time and thoughtful feedback!

---

### Official Review · Reviewer_QZfs · 2024-11-04

**Soundness:** 2
**Presentation:** 3
**Contribution:** 3
**Rating:** 6
**Confidence:** 4

**Summary:**

The paper introduces ASLDiff, a new method for finding information sources in complex networks. ASLDiff combines diffusion models with information propagation principles to accurately locate sources across different network types and patterns. It uses pre-estimated source locations as guides, a diffusion process led by these estimates, and a GCN (Graph Convolutional Network) to capture key propagation details. ASLDiff outperforms certain existing methods, achieving up to 7.5%-12.1% higher accuracy on some real-world datasets and adapting effectively to some networks and scenarios.

**Strengths:**

1. The paper combines diffusion models with principles of information propagation, offering a unique solution to the source localization problem.

2. ASLDiff leverages pre-calculated source estimations as informative priors, potentially improving efficiency and effectiveness.

3. The authors test their model on both synthetic and real-world datasets, comparing it against several state-of-the-art methods.

4. The proposed model shows promise for some real-world scenarios, such as epidemiology, and cybersecurity.

**Weaknesses:**

1. While the authors use some real-world datasets, more extensive testing on diverse real-world datasets could further validate the model's effectiveness.

2. The paper doesn't discuss the computational complexity or resource requirements of ASLDiff compared to other methods.

3. The paper does not sufficiently address the scalability of the proposed method for large networks (millions of nodes), which is essential for real-world applications. The largest tested graph contains fewer than 15K nodes.

4. ASLDiff’s performance under the SIS diffusion pattern is inconsistent. Only three datasets are used, with performance varying across datasets and metrics. For instance, there is no improvement in AC on the Net and Jazz networks, and only a marginal increase from 0.984 (GCNSI) to 0.985 on the Power network.

**Questions:**

1. How does the computational complexity of ASLDiff compare to other state-of-the-art methods, especially for large-scale networks?

2. Can the authors provide more insights into the model's performance on large real-world networks (e.g., millions of nodes)?

3.  Why the performance of ASLDiff under the SIS diffusion pattern is inconsistent?

4. Can the proposed model handle dynamic networks where the topology changes over time?

---

> ### Author Response · Authors · 2024-11-20
>
> Thanks for your constructive feedback. We have summarized your comments into the following questions. We hope the following responses can address your concerns.
>
> **1. Diversity of real-world datasets and the validation of the model's effectiveness**
>
> Thank you for this important question. We would like to address the comprehensiveness of our experimental validation from three aspects:
>
> First, our evaluation encompasses diverse datasets combining different network structures and propagation patterns. Specifically:
>
> - Three real-world networks (Jazz, Net, Power) with distinct structural properties (scale, density, etc.) as shown in the description of the networks in the appendix.
> - Three synthetic propagation patterns (SIS, IC, LT) representing different diffusion mechanisms
> - Two larger real-world networks with authentic propagation traces
>
> The dataset composition, which has a total of 11 datasets (3 networks×3 patterns + 2 real-world datasets), offers comparable or greater diversity than existing methods. For instance, the database has a comparable or greater scale to GCNSI (10 datasets), TGASI (6 datasets), and DDMSL (10 datasets).
>
> Third, we conducted extensive adaptivity experiments to validate real-world effectiveness:
>
> - Few-shot learning tests on networks with different propagation patterns(Section 5.3.1). Our model only requires at most 3% of the real dataset for fine-tuning to achieve optimal performance, demonstrating that the model can be pretrained on established propagation models for effective few-shot or zero-shot learning
> - Zero-shot transfer experiments across different network topologies(Section 5.3.2). Our model trained on synthetic networks achieves at least 86% of the performance of the one originally trained on real networks, validating the zero-shot transferability across networks
>
> These comprehensive evaluations demonstrate not only our model's effectiveness across diverse scenarios but also its strong adaptability to new, unseen situations - a critical capability for real-world applications.
>
> **2. Computational complexity of ASLDiff**
>
> We acknowledge the reviewer's concern about the complexity of our diffusion-based approach. In response, we present a detailed comparison of the computational cost of our proposed model against baselines on Twitter Dataset, as outlined in the table below:
>
> **average training time on Twitter Dataset**
>
> | Metrics  | SLVAE | GCNSI | TGASI | DDMSL | ASLDiff | ASLDiff's pretraining + few-shot|
> |----------|-------|-------|--------|--------|----------|---|
> | Avg Training Time  | ~2.5h | ~1.5h | ~2.5h | ~3h | ~3h | ~1h + ~0.5h|
>
> **average inference time per sample**
>
> | Method | LPSI | SLVAE | GCNSI | TGASI | DDMSL | ASLDiff |
> |----------|-------|-------|-------|--------|--------|----------|
> | Avg Inference Time | ~10s | ~30s | ~20s | ~20s | ~30s | ~40s |
>
> In terms of computational efficiency, ASLDiff achieves a reasonable training time, requiring less than 4 hours on a single RTX-2080 Ti GPU.
> It is notable that:
> - While our model's training duration is marginally higher due to the iterative nature of the DDPM-based denoising process, **this initial time investment can be offset by ASLDiff's crucial advantage: its few-shot and zero-shot adaptability to various networks and patterns, which significantly reduces computational resources in real-world applications.** As shown in the table, our model requires less time to pretrain on the synthetic network and finetune on the target network(correspond to Section 5.3.2) than initially training on the target network since the scale of the network in the pretrain data can be smaller.
> - We opted for DDPM as our foundation due to its classical design and proven effectiveness. It's worth highlighting that ASLDiff's architecture is fully compatible with more computationally efficient diffusion variants, such as DDIM [1], which could substantially reduce the current computational overhead. This flexibility, combined with our model's transfer capabilities, makes ASLDiff particularly resource-efficient in practical deployments. We have incorporated a detailed efficiency analysis in Appendix I of the revised manuscript.
>
> [1] Song, Jiaming, Chenlin Meng, and Stefano Ermon. "Denoising Diffusion Implicit Models." International Conference on Learning Representations. 2020.

---

> ### Author Response · Authors · 2024-11-20
>
> **3. The scalability of the proposed method**
>
> We appreciate the reviewer's concern about scalability. While our current experiments focus on networks with up to 15K nodes (similar to existing baselines), we acknowledge that scaling to million-node networks presents significant challenges and should be discussed more explicitly in the paper.
> The networks tested in our work (up to 15K nodes) align with the scale commonly studied in source localization research, as demonstrated by recent work like SLVAE, TGASI, and DDMSL. This scope represents a deliberate trade-off for learning-based methods: we prioritized developing a method that achieves higher accuracy in source localization for moderate-scale networks, rather than compromising precision to accommodate larger scales.
>
> For million-node networks, while the direct application of our current method may not be feasible, the principles we developed could inform hierarchical solutions. We envision two promising directions:
>
> - Community-based approaches that first identify candidate regions before detailed source localization
> - Multi-scale methods that progressively narrow down source locations across network hierarchies.
>
> Both these approaches would effectively reduce the scale of the network, where techniques similar to our method could potentially be applied to these smaller-scale subgraphs. This would allow leveraging our method's demonstrated strength in accurate source localization at moderate scales.
>
> We have added a detailed discussion in Appendix H about our limitations and removed potentially misleading claims about large-scale applicability.
>
> **4. The inconsistency of ASLDiff's performance under the SIS diffusion pattern**
>
> The inconsistency of ASLDiff's performance under the SIS diffusion pattern is mainly due to the variation of the datasets' scales, as well as the different focuses of metrics.
>
> **Performance varying among datasets**
>
> The main performance variations across datasets lie in the recall rate in the Jazz dataset. The recall gap is less significant than it appears because the Jazz network only has ~10 source nodes per infection scenario, so a ~10% gap only indicates 1-2 undetected source nodes. This dataset is smaller (198 nodes) compared to Net (1,589) and Power (4,941), which makes performance metrics more sensitive to individual prediction errors. We have revised the discussion in Section 5.2.1 to clarify the performance variations in the Jazz dataset.
>
> **Performance varying among metrics**
>
> Different evaluation metrics serve distinct purposes: Accuracy (AC) measures overall classification correctness across all nodes, while the F1-score balances precision and recall specifically for source identification. For source localization problems, the F1-score is more critical since AC can be misleadingly high due to the large class imbalance (very few nodes are actual sources).
>
> While ASLDiff shows modest AC improvements, it achieves significant gains in F1-score: 3.2% improvement on the Power network (0.877 vs 0.849) and 1.8% on the Net network (0.816 vs 0.801). These improvements stem from better-balanced precision and recall, meaning ASLDiff more accurately identifies true source nodes while minimizing false positives. Though we do not achieve the best F1 in Jazz, our model's gap between the best SLVAE is insignificant, as illustrated above. Our model also consistently achieves the best precision. This is particularly valuable since misidentifying source nodes (false positives) is costly in practical applications, since resources would be wasted investigating non-source nodes.
>
> To clarify this point, we have added a discussion in Section 5.2.1 explaining why the F1-score is a more meaningful metric than AC for source localization and included a demonstration of how ASLDiff achieves better precision-recall balance compared to baselines.

---

> ### Author Response · Authors · 2024-11-20
>
> **5. Can the proposed model handle dynamic networks where the topology changes over time?**
>
> This is an interesting and challenging question. Dynamic network topology adds another layer of complexity to the source localization problem, which can be categorized into two situations.
>
> First, if the dynamics of network topology are not relevant to the variation of infection states, (1)assuming that a complete historic topology variation trajectory is provided, our method could potentially be trained to predict the previous infection state based on current observations and topology, and apply this process iteratively for source localization; (2)if complete historic topology variation trajectory is not provided, it requires the model to rebuild the dynamic of network topology and localize the source simultaneously, which is out of scope of our research.
>
> Second, if the dynamics of network topology are relevant to the variation of infection states, it becomes a co-evolution[1] problem, which currently lacks theoretical basis and enough real-world data. Under the current framework of knowledge-enhanced data-driven paradigm, it is challenging to develop effective solutions to such a problem.
>
> [1] T. Gross, B. Blasius. Adaptive coevolutionary networks: a review[J]. Journal of the Royal Society Interface, 2008, 5(20):259–271**

---

> ### Author Response · Authors · 2024-11-22
>
> We appreciate your valuable time and high-quality review. We have responded to your concerns and questions about **dataset diversity, complexity, scalability, and result inconsistency**. We have also **replied to your question, demonstrating the situation and complexity the proposed model may face under dynamic network topology**.
>
> Could you please review our updates and let us know if they address your concerns? We are happy to answer any further questions.

---

> > ### Comment · Reviewer_QZfs · 2024-11-23
> > **Response to the Authors' Rebuttal**
> >
> > Thank you for the detailed response and addressing most of the concerns.
> >
> > Despite the clarifications, I find ASLDiff’s performance under the SIS diffusion pattern to be inconsistent, with notable variations across datasets and metrics (e.g., AC). While the authors have provided some insights, providing results on additional datasets could strengthen their claim and provide a more comprehensive evaluation of ASLDiff’s effectiveness.

---

> ### Author Response · Authors · 2024-11-27
>
> Thanks for your feedback. To demonstrate the consistency and robustness of ASLDiff on SIS, we conducted extensive experiments on two representative networks with distinctly different characteristics:
>
> **Table 1: Results on Digg Network with SIS Model**
>
> Digg (Rossi & Ahmed, 2015), adopted in SLVAE, is a real-world social network dataset showing voting records of stories that made it to Digg's front page in 2009. The network contains 14,511 nodes and 194,405 edges. The experiment uses the SIS propagation model with parameters:
>
> - Distribution of initial seed ratio: Uniform(0.0015, 0.015)
> - Infection probability: 0.0001
> - Recovery probability: 0.00005
>
> | Model   | F1    | RE    | PR    | AC    |
> |---------|-------|-------|-------|-------|
> | LPSI    | 0.8265| **0.8969**| 0.769 | 0.9968|
> | GCNSI   | 0.5619| 0.5043| 0.6344| 0.9643|
> | DDMSL   | 0.7586| 0.722 | 0.8001| 0.9832|
> | Ours    | **0.8403**| 0.8307| **0.852**| **0.9973**|
>
> **Table 2: Results on ego-Facebook Network with SIS Model**
>
> Ego-Facebook(J. McAuley & J. Leskovec, 2012), adopted in GCNSI, contains social circles from Facebook, consisting of 4,039 nodes and 88,233 edges. The experiment uses the SIS propagation model with parameters:
>
> - Distribution of initial seed ratio: Uniform(0.001, 0.003)
> - Infection probability: 0.0001
> - Recovery probability: 0.00002
>
> | Model   | F1    | RE    | PR    | AC    |
> |---------|-------|-------|-------|-------|
> | LPSI    | 0.7757| 0.8365| 0.7545| 0.9990|
> | GCNSI   | 0.4720| 0.4086| 0.5588| 0.9976|
> | DDMSL   | 0.5659| 0.5379| 0.6196| 0.9977|
> | Ours    | **0.8316**| **0.8570**| **0.8189**| **0.9992**|
>
> These results demonstrate strong consistency in that ASLDiff maintains superior F1, PR, and AC performance across networks with different scales and characteristics. Together with the results in the manuscript, these comprehensive evaluations strongly support ASLDiff's effectiveness under the SIS diffusion pattern.

---

> > ### Comment · Reviewer_QZfs · 2024-12-02
> >
> > Thank you for sharing the updated results. I recommend incorporating them into the paper. I have raised my score accordingly.

---

> ### Author Response · Authors · 2024-12-03
>
> Dear reviewer,
>
> Thank you for your kind feedback and suggestions. We are glad that our rebuttal has addressed your concerns and deeply appreciate the raised score. We will incorporate them into the final paper following your advice.
>
> Thank you again for your time and consideration.

---

### Author Response · Authors · 2024-11-27

Dear reviewers,

We would like to highlight again that our work offers significant practical value through its ability to handle real-world scenarios with limited domain data effectively. This capability is demonstrated through our experiments in Section 5.3, which show that ASLDiff can achieve strong few-shot and even zero-shot performance when pretrained on synthetic data. Our experiment in the digg dataset shows that our model only requires 3% of real-world data for fine-tuning to achieve optimal performance, and in the Twitter dataset, it can even perform well with no fine-tuning. This practical advantage stems from our innovative design incorporating domain knowledge into the diffusion framework, allowing the model to learn pattern-invariant features that generalize well across different propagation scenarios.

The real-world applicability is particularly valuable because there is often very limited historical data available for training in many practical situations, such as emerging disease outbreaks or novel forms of cyber attacks. ASLDiff is a practical solution that can be quickly deployed without extensive historical data collection.

---

### Author Response · Authors · 2024-12-04
**General Response**

We sincerely thank all the reviewers for their valuable and constructive comments. We greatly appreciate that the reviewers recognized most of our work. Specifically, Reviewer Qzfs, U3Da, and h6eH **acknowledged the comprehensiveness of our evaluation of the proposed method**; Reviewer NsyM agreed with **our method's few-shot learning capability and the superiority of our single-snapshot approach over multiple-snapshot methods**; Reviewer Qzfs also highlighted **the promising real-world applications in real-world scenarios such as epidemiology and cybersecurity**.

In response to key concerns, we have made significant improvements:

1. Regarding the **inconsistent performance under SIS patterns** (raised by Reviewer QZfs and h6eH), we conducted extensive experiments on two additional representative networks - Digg (14,511 nodes) and ego-Facebook (4,039 nodes). Our results demonstrate consistently superior performance across different scales, with ASLDiff achieving significant improvements in F1-scores (0.8403 and 0.8316) compared to baselines. These comprehensive evaluations strongly validate our model's effectiveness under SIS diffusion patterns. We will incorporate the result in our final paper. Please refer to our response to Reviewer QZfs for the results.

2. On **computational efficiency concerns** (raised by multiple reviewers), we presented a detailed comparison of the computational cost of our proposed model against baselines on the Twitter Dataset (please refer to our response to Reviewer QZfs). We also implemented DDIM, an efficient diffusion variant, reducing inference time by approximately 50% while maintaining performance (maximum performance drop of only 3.3%)(please refer to our response to Reviewer h6eH). The results show that our model can reduce the current computational overhead while keeping the main performance with the help of the computationally efficient diffusion variant. This flexibility, combined with our model's transfer capabilities, makes ASLDiff particularly resource-efficient in practical deployments.


3. Addressing the **practical deployment concerns** (particularly emphasized by Reviewer NsyM), we have demonstrated ASLDiff's strong capability in handling real-world scenarios with limited data in the original paper. Our experiments show that ASLDiff requires only 3% of real-world data for fine-tuning on the Digg dataset to achieve optimal performance, and can achieve effective zero-shot performance on the Twitter dataset. This efficiency stems from the rich synthetic pretraining data incorporating diverse propagation patterns and the universal property guidance through prior-guided diffusion, which captures pattern-invariant properties while maintaining adaptability to pattern-specific features (see experimental results from our response to Reviewer h6eH). We agree that the few-shot learning capability is a key contribution that deserves more thorough elaboration. We have followed Reviewer NsyM's suggestion to add detailed discussions on the model's potential few-shot learning capabilities in the method session and the result section.

We conduct additional experiments and revisions on the original manuscript to address the concerns of reviewers. A brief summary of the results includes:

1. (Reviewer h6eH & QZfs) We conducted extensive experiments on two new networks (Digg and ego-Facebook) with SIS diffusion pattern, showing consistent superior performance. We also added comprehensive runtime comparisons across different models
2. (Reviewer h6eH) We performed experiments with the DDIM variant, reducing steps from 500 to 250, demonstrating maintained performance while reducing inference time by ~50%. We also merged Sections 2.1 and 3.2 to reduce redundancy, added a detailed discussion in Appendix H about our limitations, and removed potentially misleading claims about large-scale applicability. Furthermore, we also conducted additional experiments to better investigate how ASLDiff models different patterns.
3. (Reviewer QZfs) We revised the discussion in Section 5.2.1 to clarify the performance variations in the Jazz dataset.
4. (Reviewer NsyM) We enhanced the methodology section with a detailed few-shot learning discussion and expanded Section 5.3 with a comprehensive few-shot learning results analysis.
5. (Reviewer U3Da) We enhanced our manuscript with more detailed explanations of our innovative aspects of the technical details and improved the clarity of our framework diagrams in the revised manuscript.

---

### Meta-Review · Area_Chair_83My · 2024-12-23

**Metareview:**

The paper introduces a method to solve the source localization problem based on a diffusion model and GNNs. In terms of overall ranking, a majority of reviewers were mildly negative about the paper and one reviewer was mildly positive. The main concerns had to do with the experimental evaluation, particularly in terms of generalization, scalability and presentation/motivation. As a consequence, I am unable to recommend acceptance.

**Additional Comments On Reviewer Discussion:**

The authors put a significant effort in their rebuttal, including running additional experiments, however, a majority of the reviewers were not persuaded to support acceptance.

---

### Decision · Program_Chairs · 2025-01-22

Reject